# Ground-Truth Subgraphs for Better Training and Evaluation of Knowledge Graph Augmented LLMs

## Abstract

Retrieval of information from graph-structured knowledge bases represents a promising direction for improving the factuality of LLMs. While various solutions have been proposed, a comparison of methods is difficult due to the lack of challenging QA datasets with ground-truth targets for graph retrieval. We present SynthKGQA, an LLM-powered framework for generating high-quality Knowledge Graph Question Answering datasets from any Knowledge Graph, providing the full set of ground-truth facts in the KG to reason over questions. To demonstrate its utility, we apply SynthKGQA to Wikidata to generate GTSQA. This new dataset is specifically designed to test zero-shot generalization with respect to unseen graph structures and relation types, enabling us to analyze the abilities and limitations of SOTA graph retrieval approaches at an unprecedented level of granularity. We also show that KG retrievers trained on GTSQA can transfer to human-curated benchmarks, and that the ground-truth subgraphs produced by SynthKGQA provide a better training supervision signal than previously-used heuristics.

## 1 Introduction

Despite significant advances over the years, Large Language Models (LLMs) are still unreliable when asked to provide factual information, as hallucinations remain one of the central problems for LLM applications (Huang et al., 2025). The predominant solution to improving LLM trustworthiness is Retrieval Augmented Generation (RAG), where information pertinent to the query is retrieved from a corpus of knowledge and added to the prompt (Lewis et al., 2020; Borgeaud et al., 2022; Izacard et al., 2023). While RAG traditionally retrieves information from documents, another important use case is retrieval from graph structured repositories such as Knowledge Graphs (KGs). KGs encode facts as (subject, predicate, object) triples and are a highly efficient solution to store *relational* information while also being easier to maintain, update, and fact-check compared to textual documents.

KG-augmented LLMs are evaluated on Knowledge Graph Question Answering (KGQA) benchmarks, where supporting facts need to be extracted from the KG and combined to produce the answer. While many KGQA datasets have appeared over the years (see Peng et al. (2025) for a survey), little attention has been paid to their quality, reliability and limitations. The recent study by Zhang et al. (2025) has estimated the degree of factual correctness of the questions in widely-used benchmarks, such as WebQSP (Yih et al., 2016), CWQ (Talmor & Berant, 2018) and GrailQA (Gu et al., 2021), to be only between 30 and 60%. Moreover, several benchmarks are no longer challenging for SOTA KG-augmented LLMs and are now close to being saturated, limiting their usefulness. Even for datasets like ComplexWebQuestions (CWQ), which include multi-hop questions, it is impossible to provide an actual measure of question complexity due to the lack of *ground-truth answer subgraphs*, the golden targets for retrieval. This also means that KG-RAG retrievers cannot be evaluated on their own, but only end-to-end, by looking at the final answer provided by the LLM – which, since different solutions use different LLMs and prompting schemes, results in noisy comparisons. Moreover, for KG retrievers that require training, the ground-truth answer subgraph is indispensable to provide supervision signal: when using existing datasets, it has to be approximated, typically by means of the set of shortest paths from seed entities to answer nodes. As we show in Section 6, this is often a bad approximation, which penalizes the final quality of the retriever. Furthermore, most KGQA benchmarks

are from the previous decade and based on the now discontinued Freebase KG (Bollacker et al., 2008), thus containing outdated facts and answers that might be in conflict with the more recent information available to LLMs. These datasets have also likely been seen during pretraining of recent LLMs (Ding et al., 2024), leading to data leakage. Question variety, in terms of KG relation and entity types used for the grounding facts, is also typically limited, as questions are either manually (through crowd-sourcing) or procedurally (through logical query manipulation) generated from predefined query templates, often resulting in unnatural, or ambiguously phrased, natural language questions (see Zhang et al. (2025) for a more detailed discussion of limitations and pitfalls).

All these challenges can be tackled by leveraging the abilities of frontier LLMs within a controlled pipeline for the creation of KGQA datasets, removing the need for a human in the loop. We release SynthKGQA[1] (Figure 1), a framework for generating large KGQA datasets from any KG. It provides high-quality, diverse questions with procedurally-verified ground-truth answer subgraphs and SPARQL queries, allowing to easily update the dataset whenever the underlying KG is modified. SynthKGQA improves upon previous solutions for the generation of synthetic KGQA datasets, tackling major limitations. By leveraging it, we achieve the following contributions:

- *Ground-Truth Subgraphs for Question Answering* (GTSQA)[1], a challenging new dataset with 32,099 questions spanning 27 different structures for the ground-truth answer subgraph, grounded in the regularly-updated Wikidata KG (Vrandečić & Krötzsch, 2014), and specifically designed to test generalization abilities of KG-RAG models. We also show that KG retrievers trained on GTSQA exhibit strong 0-shot transfer performance to human-curated KGQA datasets, making GTSQA not just a benchmarking dataset, but also a precious training resource.

- A comprehensive benchmark of SOTA LLMs and KG-augmented LLMs on GTSQA, providing new insights on retrieval abilities and limitations of these models on different structures for the answer subgraph, and different levels of zero-shot generalization.

- The first numerical quantification and extensive analysis of the benefits of using the ground-truth answer subgraph, instead of shortest paths from seed to answer nodes, as supervision signal for training KG retrievers.

## 2 Related work

Traditionally, models for KGQA have been evaluated on datasets built using a combination of handcrafted templates (for questions and logic queries) and human annotation, often from the Freebase KG (Bollacker et al., 2008). WebQuestions (Berant et al., 2013) used autocompletions from the Google Suggest API on Freebase entity labels, with answers provided by human annotators – a process resulting prevalently in 1-hop questions. Many datasets are derived from WebQuestions, by filtering and refining it (WebQSP (Yih et al., 2016)) or extending/combining its SPARQL queries to obtain more complex questions (CWQ Talmor & Berant (2018)). With LC-QuAD (Trivedi et al., 2017), more structured, and partially automated, pipelines started to appear, mapping logical queries to natural language using question templates and crowd-sourced paraphrasing. Moreover, after Freebase was discontinued in 2016, new datasets shifted to using different KGs, mainly DBpedia (e.g., QALD-9 (Usbeck et al., 2018)) and Wikidata (e.g., LC-QuAD 2.0 (Dubey et al., 2019)), with parallel efforts to migrate to these KGs previous datasets based on Freebase (Azmy et al., 2018). New interest developed also in testing generalization abilities of KGQA models (GrailQA (Gu et al., 2021)) and better quantifying the complexity of questions (GrailQA++ (Dutt et al., 2023)).

Only very recently Dammu et al. (2025) and Zhang et al. (2025) started exploring the use of LLMs in the construction of KGQA datasets. Despite the implementation of pipelines to detect and discard bad data, these approaches still suffer from hallucinations leading to incorrect question-answer pairs (Zhang et al., 2025). Also, fundamental information such as the complete set of entities mentioned in each question, or fine-grained measures of question complexity or generalization skills required to answer, are not fully tracked.

---

[1]See supplementary material; links to github repository and dataset to be added after double-blind review.

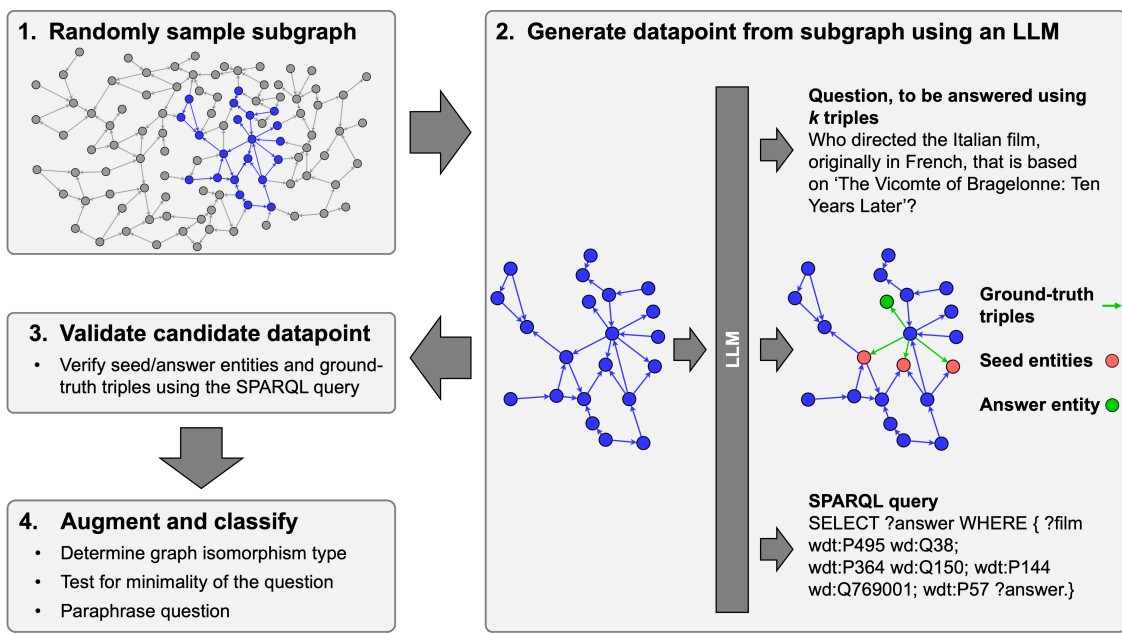

Figure 1: The steps performed by SynthKGQA to generate and validate questions and ground-truth subgraphs from an arbitrary Knowledge Graph.

Crucially, also due to such shortcomings, very limited analysis of SOTA KG retrievers has been conducted on these new datasets, with benchmarks only looking at the performance of out-of-the-box LLMs (Dammu et al., 2025) or proving unreliable due to severe issues in the choice of seed entities in the KG retrieval step (Zhang et al., 2025). See Section 3 and Appendix F for an extensive discussion.

## 3   The SynthKGQA Framework

KGs store facts as labeled directed edges between *entities* in a set $\mathcal{E}$ (the nodes/vertices of the KG), where edge labels are drawn from a set $\mathcal{R}$ of predicates, or *relation types*. Thus, an edge is represented as a subject-predicate-object triple $(h, r, t)$, with $h, t \in \mathcal{E}$ and $r \in \mathcal{R}$; we denote by $\mathcal{T}$ the set of triples in the KG. KGQA questions require retrieving a subgraph $\mathcal{G} \subset \mathcal{T}$ and reason over it to produce an answer. We assume that, together with the question $q$, the set of *seed entities* $\mathcal{S} \subset \mathcal{E}$ is provided, i.e., the set of entities that are explicitly mentioned in the question. Extracting seed entities from natural language text is an orthogonal task referred to as Named Entity Recognition, or Entity Linking, which is in itself an object of extensive research (Alam et al., 2022; Keraghel et al., 2024).

Here we outline SynthKGQA, our proposed framework, which can be applied to any KG in order to construct high-quality synthetic data for KGQA (Figure 1). Powered by an LLM, SynthKGQA starts from randomly sampled subgraphs $\mathcal{Q} \subset \mathcal{T}$ and – based on these – generates candidate datapoints consisting of a *question*, the *ground-truth subgraph* needed to answer it, the *answer*, the *seed entities*, and a *SPARQL query* that encodes the natural-language question $q$ in logical form. These datapoints get validated by executing the SPARQL query against the knowledge base, to retrieve the full set of answers and reasoning paths. To increase diversity of the generated data, we ask a different LLM to paraphrase $q$ in more natural terms, and, by using the ground-truth answer subgraph and SPARQL query, we provide additional information on each valid datapoint:

- the *graph isomorphism type*, the structure of the ground-truth answer subgraph up to isomorphism (Appendix A.2), as a measure of question complexity;

- *question minimality*, a quantification of whether the provided set of seed nodes is *minimal*, or if the question can be answered by using only a subset $\mathcal{S}' \subsetneq \mathcal{S}$ (see Appendix A.3).

Table 1: Overall view of GTSQA.

|  | Train | Test | All |
| --- | --- | --- | --- |
| # questions | 30,477 | 1622 | 32,099 |
| # unique relation types | 200 | 362 | 368 |
| # unique entities | 64,435 | 5665 | 68,520 |
| # unique graph isomorphisms | 19 | 14 | 27 |
| # non-minimal questions | 7852 | 0 | 7852 |
| avg # seed entities | 1.65 | 1.92 | 1.66 |
| avg # hops | 1.48 | 2.02 | 1.5 |
| avg # answers | 1.54 | 1.28 | 1.53 |
| avg # ground-truth edges | 2.13 | 3.03 | 2.17 |

For KG retriever evaluation, we make the assumption that the answer(s) to the question are entities in a set $\mathcal{A} \subset \mathcal{E}$. However, SynthKGQA can be extended beyond this standard setting. As we show in Appendix A.5, for instance, it can also produce questions requiring aggregations, filtering, and negations in the SPARQL query without any additional overhead. More details on the data generation and validation steps of SynthKGQA are provided in Appendix A.1.

In contrast to classical KGQA datasets that rely on handcrafted templates for logic queries (Section 2), the SPARQL queries in SynthKGQA are entirely generated by the LLM (while the user retains a fine-grained control over the logic operators and aggregation types to be used by specifying them in the prompt). This significantly increases the variety of topological structures for the logical paths connecting the seed entities to the answer. Moreover, while classical datasets use such queries to construct questions, we use them instead to procedurally validate the quality of the synthetic data generated by our pipeline, in order to catch LLM hallucinations (an issue still present in recent attempts to use LLMs to generate synthetic KGQA data, such as Dammu et al. (2025)). While the KGQAGen pipeline from Zhang et al. (2025) also involves executing the SPARQL queries, it exhibits biases in the way questions are constructed from subgraphs (Appendix F). Moreover, it does not track entities mentioned in questions, nor has a way of measuring question complexity or minimality. To our knowledge, SynthKGQA is the first framework providing the full data required to effectively train and benchmark KG retrievers, while supporting cheap regeneration of datapoints whenever the underlying KG is updated (simply by rerunning the SPARQL queries).

In Section 4 we apply SynthKGQA to the Wikidata KG (Vrandečić & Krötzsch, 2014) and present benchmarking results in Section 5. Notably, though, the SynthKGQA framework is KG-agnostic and can be applied to any custom KG, even domain-specific ones: we provide evidence of this in Appendix E, where we conduct a small-scale experiment on constructing KGQA data from a biomedical KG.

## 4 The GTSQA Dataset

Using the SynthKGQA framework and GPT-4.1 (OpenAI, 2023) as LLM, we construct *Ground-Truth Subgraphs for Question Answering* (GTSQA), a synthetic KGQA dataset grounded in the Wikidata KG (Vrandečić & Krötzsch, 2014), with 30,477 questions in the train set and 1622 questions in the test set. Statistics on the dataset are summarized in Table 1, with more details provided in Appendix B.

GTSQA supports retrieval either from the full Wikidata, or from ogbl-wikikg2 (Hu et al., 2020), which was employed to sample the seed graphs $\mathcal{Q}$. We prompt SynthKGQA's LLM to construct questions with at most 6 edges in the ground-truth answer subgraph, as we observe that – beyond this point – questions tend to become overly-convoluted and unnatural to formulate in words. We also require the ground-truth answer subgraph to be a tree, i.e., connected and acyclic, with all leaves being seed entities or the answer node. This is to avoid the occurrence of circular questions, where the answer is mentioned verbatim in the question (a problem observed in 9.4% of the test questions in KGQAGen-10k (Zhang et al., 2025), but never occurring in GTSQA). The dataset covers 27 different graph isomorphism types for the ground-truth answer subgraph, with questions involving up to 5 hops and up to 5 different seed entities. Full statistics on the

connectivity patterns of seed and answer nodes, and relative frequencies in the dataset, are provided in Table 5 and Appendix B. As shown in Figure 6, approximately 70% of the test questions are multi-seed (two or more seed entities) and 67.9% are multi-hop (with 30.8% requiring $\geq 3$ hops). This variety of graph types makes GTSQA much more challenging than commonly-used benchmarks for KGQA (only 34.5% of the test questions in WebQSP (Yih et al., 2016) require more than 1 hop, and none require more than 2). Moreover, the test set of GTSQA has the unique property of containing only minimal questions (as defined in Section 3), where the full set of seed entities is necessary to answer. Since the ground-truth answer subgraph is a tree, this ensures that the full set of edges is required and the provided classification of graph isomorphism types is truly reflective of the question complexity.

A notable feature of GTSQA is that the train-test split is designed to test zero-shot generalization abilities of KG retrievers (in a spirit similar to the construction of GrailQA (Gu et al., 2021) and GrailQA++ (Dutt et al., 2023)). We ensure that the answer nodes for all test questions never appear as an answer in training questions. Moreover, 47.3% of the answer subgraphs in the test set have as isomorphism type one of 8 classes that are not included in the train set (see Table 5), while 37.2% of test questions require reasoning over edges whose relation type is not seen in the train set (see Figure 5 for the distribution of relation types). We also make sure that, for each graph isomorphism type in the test set, the applicable categories (in-distribution, unseen graph type, unseen relation type) always contain at least 45 different questions, to enable a statistically meaningful study of KG retrievers' performance at this unprecedented granularity level (as we do in Section 5).

We perform a final filtering step on the test set of GTSQA to ensure its value and reliability as a benchmark for KG retrievers: we retain only questions that can be answered correctly by an advanced LLM (GPT-4o-mini) when augmenting the prompt with the ground-truth answer subgraph provided in the dataset, in a consistent way (two successes out of two tries). This is to sanity check that the ground-truth answer subgraph is indeed a valid golden target for retrieval. We found that only 0.47% of the generated data failed this test (and was therefore removed), which confirms the very high quality of the data produced by SynthKGQA (compared, for instance, to KGQAGen-10k, where this failure rate is reported at 10.38% with the more capable GPT-4o, and failure causes remain unclear).

## 5 Benchmarks

We benchmark SOTA KG-RAG models on GTSQA. As one of the key features of our dataset is that it provides ground-truth answer subgraphs for each question, we can evaluate not just the quality of the models' final answer, but also the quality of the retrieved subgraph.

### 5.1 Experimental setting

We consider four categories of models for QA (specifications in Appendix C.2). 1) *LLM-only*: as a baseline, we take frontier commercial LLMs without any external augmentation pipelines, such as GPT-4.1, GPT-4o-mini (OpenAI, 2023), GPT-5-mini (OpenAI, 2025), Ministral-8B-Instruct, Mistral-Large-2.1 (Mistral AI, 2024) and LLaMA-3.1-8B-Instruct (AI@Meta, 2024). 2) *KG agents*: training-free models using out-of-the-box LLMs to explore the KG starting from the seed entities. They decide on their own, over multiple steps, what neighbors to explore and when to stop. As representatives, we consider Think-on-Graph (ToG; Sun et al. (2024)) and Plan-on-Graph (PoG; Chen et al. (2024)). 3) *Path-based retrievers*: models trained to predict the paths originating from the seed entities and leading to the answer node. Examples are SR (Zhang et al., 2022), Reasoning on Graphs (RoG; Luo et al. (2024)) and Graph-Constrained Reasoning (GCR; Luo et al. (2025)). 4) *All-at-once retrievers*: models trained to score all edges in a large neighborhood of the seed entities in a single pass, and then retrieve the most relevant ones. We consider SubgraphRAG (Li et al., 2025) as a representative.

For all models in categories 2), 3), 4), we use the same LLM (GPT-5-mini) to perform the final reasoning on the retrieved subgraph, in order to make the benchmark as fair as possible and place the main focus on the quality of the retrieved subgraph. We use ogbl-wikikg2 as KG for retrieval, to reduce computational complexity (see also Appendix C.1). As common in KGQA benchmarking (Sun et al., 2024; Luo et al., 2024),

Table 2: Benchmark of LLMs and KG-RAG models on GTSQA.

| Category | Model | EM | | Ground-truth triples | | | Answer nodes | | # triples |
| | | Hits | Recall | Recall | Precision | F1 | Hits | Recall | |
| --- | --- | --- | --- | --- | --- | --- | --- | --- | --- |
| LLM-only | Ministral-8B-Instruct | 10.73 | 10.16 | - | - | - | - | - | - |
| | LLama-3.1-8B-Instruct | 17.11 | 16.33 | - | - | - | - | - | - |
| | GPT-4o-mini | 20.90 | 19.93 | - | - | - | - | - | - |
| | Mistral-Large-2.1 | 23.61 | 22.85 | - | - | - | - | - | - |
| | GPT-5-mini | 31.44 | 30.20 | - | - | - | - | - | - |
| | GPT-4.1 | 33.97 | 32.83 | - | - | - | - | - | - |
| KG agent | PoG | 41.74 | 40.32 | 36.07 | 37.14 | 34.07 | 35.76 | 33.56 | 3.72 |
| | ToG | 53.02 | 51.19 | 32.78 | 7.00 | 10.80 | 36.13 | 35.06 | 8.00 |
| Path-based | SR | 51.75 | 49.75 | 30.22 | 3.44 | 5.69 | 50.25 | 49.39 | 72.94 |
| | GCR | 56.58 | 54.59 | 40.71 | 27.21 | 29.83 | 47.10 | 45.53 | 6.54 |
| | RoG | 67.47 | 65.09 | 54.69 | 24.04 | 27.00 | 72.91 | 71.84 | 72.15 |
| All-at-once | SubgraphRAG (200) | 78.24 | 74.28 | 79.09 | 1.29 | 2.53 | 85.33 | 84.36 | 199.61 |

we evaluate quality of model responses by reporting on Hits (at least one correct answer predicted) and Recall of correct answers, using Exact Match (EM) between the labels of answer nodes and the model output. Moreover, as made possible by GTSQA, we analyze Recall, Precision and F1 score of ground-truth triples in the subgraph retrieved by the model, in addition to Hits and Recall of answer nodes.

## 5.2 Results

**GTSQA benchmarks**  A detailed comparison of the performance of different models on GTSQA is provided in Table 2.

Pure LLMs show limited accuracy with an EM Hits score of at most 33.97, confirming the challenging nature of this new benchmark for models that solely rely on their internal knowledge. Even SOTA KG-RAG models – while performing better than pure LLMs – struggle with the task. Trainable subgraph retrievers outperform KG agents, highlighting the importance of fine-tuning models on the target KG. Among trainable retrievers, SubgraphRAG (with 200 retrieved triples) achieves the best results, surpassing all path-based retrievers, which tend to retrieve smaller subgraphs. While all retrievers exhibit limited F1 scores of retrieved triples ($< 30\%$), the evidence suggests that increasing the recall of ground-truth triples is more beneficial than increasing precision, especially when the final reasoning is performed by more capable LLMs, better at dealing with long context, as we show in the comparison in Figure 14. For more compact LLMs, however, retrieval precision should not be disregarded. Also note that the recall of ground-truth triples is a much stronger predictor of final model accuracy than the recall of answer nodes (Figure 13), proving the value of working with datasets like GTSQA that provide the full ground-truth answer subgraph.

**Retrieval performance by graph type**  Breaking down model performance by graph isomorphism types (Figures 2 and 10), we find that all models, especially KG agents, have very limited accuracy on questions that require intersecting paths from 3 or more seed entities, even if the answer is only one hop away (e.g., graph isomorphisms (1)(1)(1) and (1)(1)(1)(1)). In this setting, the base LLM on its own also fails consistently. The poor EM is explained by a low recall of ground-truth triples, highlighting a widespread inefficiency in properly expanding and coordinating the search from all seed entities. Indeed, especially for KG agents and path-based methods, the recall of ground-truth triples is on average much lower in multi-seed questions (Figure 10), leading to a worse EM, despite the fact that the answer node is often contained in the retrieved subgraph (Figure 11). This is a consequence of the fact that these models sample paths from distinct seed nodes in a mostly independent way, and therefore struggle with prioritizing common neighbors of the seed nodes that satisfy at the same time all conditions in the query. GTSQA is the first benchmark to make this failure mode visible, by providing all ground-truth triples and by enforcing that all of them are required to answer the question (minimality). We show more examples of failure cases of retrievers in Appendix D.

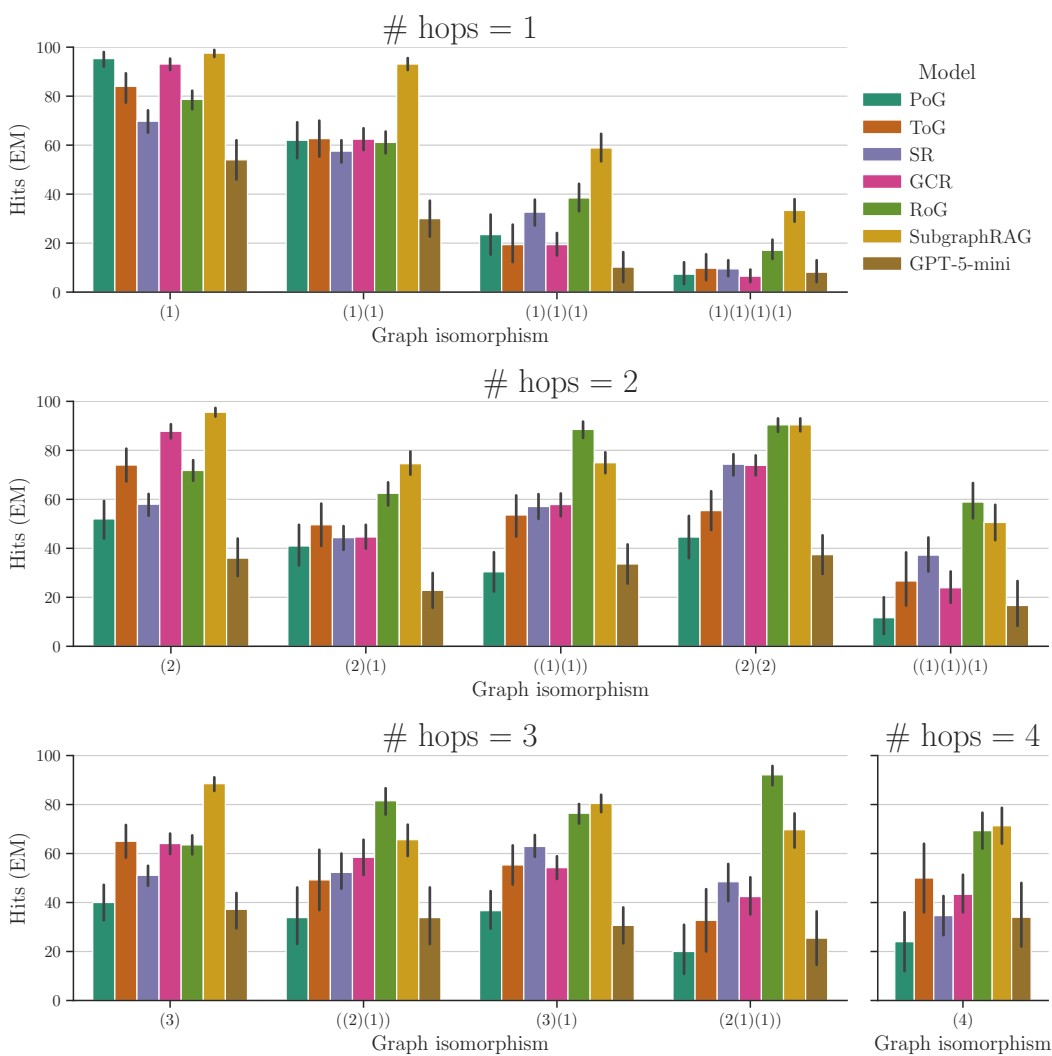

Figure 2: Hits (EM) of KG-RAG models by graph isomorphism type, compared to the baseline (GPT-5-mini, no RAG). Isomorphism types are grouped by the maximum number of hops; inside each subplot, moving from left to right corresponds to an increase in the total number of edges in the ground-truth answer subgraph.

Performance also depends on the number of hops between seed entities and the answer. Comparing KG agents, we observe that PoG matches or outperforms ToG for single-hop questions, but performs severely worse on multi-hop questions, even on simpler graph isomorphism types that involve a single seed entity (e.g., (2) and (3)). This is due to a reduced recall of ground-truth triples (Figure 10). An empirical inspection of the retrieved subgraphs indicates that PoG often prematurely stops the exploration of the KG, over-confidently (and incorrectly, in most cases) believing it has retrieved enough information to answer (see example in Appendix D). We also find a poor performance of GCR, especially when compared to its predecessor RoG, across all graph isomorphism types requiring more than 2 hops from the seed entities. This can mostly be attributed to the implementation restrictions and choices explained in Appendix C.2. Indeed, for these questions we observe that answer node recall drops much more than ground-truth triple recall (Figure 11), because only paths of up to 2 hops from the seed entities can be retrieved. However, GCR also performs worse than RoG on some questions with only 2 hops required, such as graph isomorphisms ((1)(1)) and (2)(1). Besides GTSQA being less saturated than WebQSP and CWQ, we also observe significant differences in the ranking of KG retrievers across these benchmarks (Table 7). We discuss this in Appendix C.3 based on the novel insights presented above on the limitations of the evaluated models.

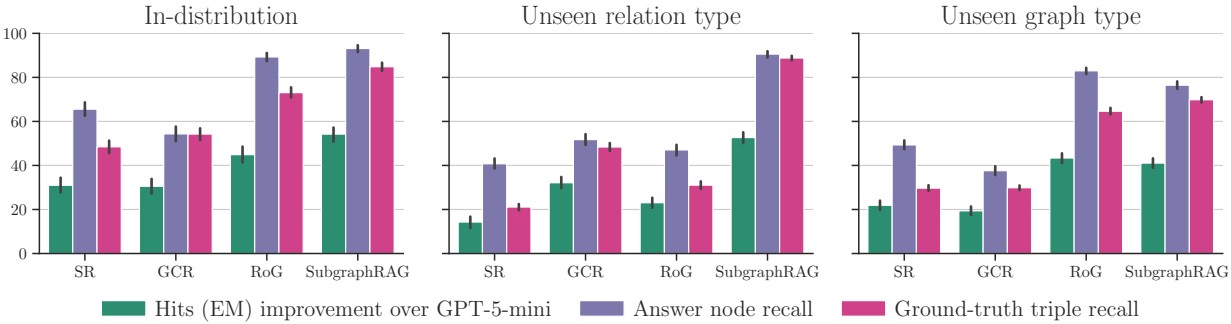

Figure 3: Generalization abilities of trainable KG-RAG models. We measure EM performance in terms of difference with the EM of the baseline (GPT-5-mini, no RAG).

Table 3: Cross-dataset transfer from GTSQA to Mintaka. We report Hits (EM) on the Mintaka test split.

|  | 0-shot | few-shot | full training |
|---|---|---|---|
| GPT-5-mini | 79.12 | - | - |
| RoG | 79.69 | 79.39 | 79.89 |
| SubgraphRAG | 85.81 | 86.02 | 86.12 |

**Out-of-distribution generalization of KG retrievers**   We investigate the zero-shot generalization abilities of trainable retrievers. When a question presents a relation or a graph isomorphism type that has not been seen during training, the improvements over the LLM baseline are smaller than for in-distribution questions (Figures 3 and 11). In these settings, we also notice stronger differences between different KG-RAG models. In particular, all path-based retrievers struggle with unseen relation types. However, GCR is much more robust than SR and RoG for this type of generalization, with significantly higher recall of ground-truth triples and EM, as its path predictions are grounded in the KG and hence less dependent on the relation types seen during training. On the other hand, the all-at-once retriever SubgraphRAG maintains strong predictive power on unseen relation types, but underperforms RoG across the majority of unseen graph isomorphism types (Figure 11), with a substantial drop in recall of answer nodes, which, combined with the low precision of retrieved triples typical of this method, results in a poor EM. These fundamental differences between path-based and all-at-once retrievers match intuition: path-based methods learn to predict the sequence of relation types that connect seed entities and answer, ignoring the global structure of the answer subgraph, whereas all-at-once retrievers like SubgraphRAG mostly leverage the graph structure during their training. SubgraphRAG especially struggles with isomorphism types requiring additional projections after an intersection (namely, $((1)(1))$, $(2)(1)(1))$, $((2)(1))$, $((1)(1))(1)$, Figure 11), a property foreign to the subgraphs in the train set – highlighting the lack of robustness of this retriever with respect to new patterns in the logical query (examples in Appendix D).

**Cross-dataset transfer**   We investigate whether KG retrievers trained on GTSQA are able to transfer beyond the dataset itself, by testing them on Mintaka (Sen et al., 2022), a well-established human-curated benchmark for retrieval from Wikidata. We adopt the same approach from Gu et al. (2021), comparing three training regimes: 0-shot (train on GTSQA, test on Mintaka), few-shot (train on GTSQA, finetune on 10% of the Mintaka training set, test on Mintaka) and full-training (train and test on Mintaka). As shown in Table 3, even in the 0-shot regime, SOTA models RoG and SubgraphRAG achieve a performance comparable to their full-training counterparts. SubgraphRAG also substantially improves over the GPT-5-mini-only baseline, proving transferability of GTSQA-trained retrievers to other datasets. This demonstrates that the distribution of questions in the SynthKGQA-generated GTSQA is well-aligned with human-curated datasets, suggesting that GTSQA constitutes a useful pre-training resource for KG retrievers.

# 6 Ground-Truth Subgraphs to Train Better KG Retrievers

In the absence of ground-truth answer subgraphs in previous datasets, KG retrievers have been trained using all shortest paths between seed and answer nodes as supervision signal (Zhang et al., 2022; Luo et al., 2024; Li et al., 2025; Luo et al., 2025; Mavromatis & Karypis, 2025). The ground-truth subgraphs in GTSQA enable us to address the following questions: 1) How much overlap is there between shortest paths and ground-truth subgraphs? 2) Are LLMs still able to answer correctly if augmented with shortest-paths between seed nodes and answer node? 3) Is using ground-truth subgraphs as supervision signal for training KG retrievers better than using shortest paths?

**Q1: Path overlap**   Paths of minimal length connecting seed nodes to answer nodes are not guaranteed to be the correct paths required to answer the question. For example, the question "What is the canonization status of Gregory the Illuminator's grandchild?" requires retrieving the 3-hop path *Gregory the Illuminator → child → St. Vrtanes I → child → St. Husik I → canonization status → saint*. However, in Wikidata, the shortest path connecting the seed node (*Gregory the Illuminator*) to the answer node (*saint*) consists of a single edge: *Gregory the Illuminator → canonization status → saint*, which clearly provides the wrong reasoning. We refer to similar cases, where the minimal length of paths between seed and answer node is strictly smaller than the length of the path in the ground-truth subgraph, as *shortcuts*. A second source of problems arises from *parallel* (shortest) *paths*: as the distance between the seed and answer node increases, the number of distinct paths of minimal length connecting them is expected to grow exponentially, but only one of them (or none, if there are shortcuts) is a ground-truth path. We provide statistics on the occurrence of shortcuts and parallel paths in GTSQA in Figure 9; their combined effect, as documented in Table 6, strongly reduces the overlap between ground-truth and shortest paths for questions requiring multiple hops. For 4-hop questions, on average, only 13% of the triples along shortest paths are contained in the ground-truth answer subgraph and more than 72% of the ground-truth triples do not lie on paths of minimal length.

**Q2: RAG with shortest paths**   There are cases where following shortcuts or parallel paths, rather than the ground-truth path that we provide in the dataset, can still produce valid reasoning. For instance, the question "Which country is home to the administrative region that includes Nieuw-Weerdinge?", which comes with the 2-hop ground-truth path *Nieuw-Weerdinge → located in the administrative territorial entity → Emmen → country → Netherlands*, can be answered equally well with the shortcut *Nieuw-Weerdinge → country → Netherlands*. Similarly, Wikidata contains pairs of inverse relation types (e.g., *child* and *father*) that give rise to parallel (undirected) paths encoding the same semantic information. We therefore look at the performance of GPT-4o-mini on the test set of GTSQA, when augmenting the prompt with all KG triples on the shortest paths between seed and answer nodes. As shown in Figure 4, there is a strong positive correlation between Hits (EM) and the fraction of ground-truth triples that are contained in the set of shortest path triples. Both metrics degrade sharply with increasing distance between seed and answer node. Note that we already adjust for any spurious effects of the number of hops on EM, since by construction GPT-4o-mini is able to answer all questions correctly when instead we augment the prompt with the triples in the ground-truth answer subgraph. This confirms that shortest paths, even though they end at the correct answer node, often fail to capture the full information required to reason over multi-hop questions.

**Q3: Shortest paths vs ground-truth subgraphs for training**   While the experiments in the previous paragraphs prove that shortest path triples are not as good a target for retrieval as the ground-truth triples provided by SynthKGQA, it remains to understand whether using them as supervision signal for training subgraph retrievers is equally sub-optimal. We consider three retrievers that use path information for training, namely SR (Zhang et al., 2022), RoG (Luo et al., 2024) and SubgraphRAG (Li et al., 2025), train them on GTSQA using either the shortest paths or the ground-truth paths between seed and answer nodes as supervision signal, and then compare the performance on the test set.

As reported in Table 4, models trained on the ground-truth subgraphs have EM Hits scores from 4% to 14% higher than their counterparts trained on shortest paths, due to improved recall (up to +27%) and precision (up to +141%) of ground-truth triples in the retrieved subgraphs. The improvements are clear and statistically significant for SubgraphRAG (independently of the number of retrieved triples, Figure 14), but even more striking for path-based methods. As shown in Figure 12, RoG is the model whose quality of

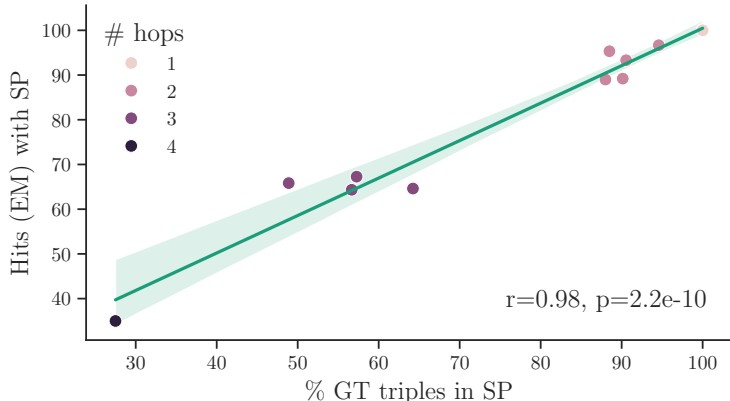

Figure 4: Correlation between the percentage of ground-truth (GT) triples contained in the set of shortest paths (SP) from seed to answer nodes, and EM Hits of GPT-4o-mini when augmented with all SP triples. We display the Pearson correlation coefficient and associated p-value; each dot represents a different isomorphism type of ground-truth answer subgraph in the test set of GTSQA: they are clearly clustered based on the (maximum) number of hops.

Table 4: Improvements in predictive statistics of models trained on ground-truth answer subgraphs (GT), compared to models trained on the shortest paths between seed and answer nodes (SP). For all models, results are averaged over three distinct runs for each of the two training regimes. We also report (between brackets) the metric's % variation and the p-value of the null hypothesis that the mean of distribution of SP scores is larger or equal than the one for GT scores (two-sample, one-sided t-test), showing statistical significance.

| Model | Hits (EM) | | Recall (GT triples) | | Precision (GT triples) | | Recall (answer nodes) | |
|---|---|---|---|---|---|---|---|---|
| | SP | GT | SP | GT | SP | GT | SP | GT |
| SR | 45.33 | 51.75 (+14%; 1.2e-10) | 23.74 | 30.22 (+27%; 6.2e-22) | 3.84 | 3.44 (-10%; 0.99) | 36.36 | 49.39 (+36%; 1.0e-39) |
| RoG | 64.22 | 67.47 (+5%; 3.6e-4) | 46.38 | 54.69 (+18%; 4.6e-26) | 10.00 | 24.04 (+141%; 0.0) | 70.68 | 71.84 (+2%; 4.3e-2) |
| SubgraphRAG | 75.11 | 78.24 (+4%; 1.3e-4) | 75.39 | 79.09 (+5%; 1.7e-9) | 1.21 | 1.29 (+6%; 4.0e-7) | 80.46 | 84.36 (+5%; 1.4e-7) |

retrieved subgraphs benefits more from eliminating the noise in the training data, with precision increasing by more than 9× for 4-hops questions. Similarly, SR experiences a major boost in answer node recall (more than 2× for 4-hops questions). While for 1-hop questions the differences in all statistics are negligible, the gap sharply increases for questions requiring multiple hops (where, as we have shown, the shortest paths diverge more appreciably from the ground-truth subgraphs). We also note that training on GT triples (rather than shortest paths) proves to be more and more beneficial to the final predictive accuracy as the reasoning capabilities of the LLM increase (Figure 14).

All this provides strong evidence for the importance of leveraging ground-truth paths to train KG retrievers, and consequently highlights the value of SynthKGQA as a resource to build datasets to train better models for complex KGQA.

## 7 Conclusions

In this work, we detailed the applications of SynthKGQA, a new framework for constructing large-scale, high-quality synthetic KGQA datasets from arbitrary KGs, with ground-truth answer subgraphs and SPARQL queries. We released GTSQA, a multi-hop, multi-seed dataset with 30k+ questions based on Wikidata, designed to test how KG-RAG models generalize to unseen answer graph structures and relation types. Our benchmarks show that SOTA KG-RAG models struggle on GTSQA, due to poor retrieval performance that

affects especially questions with multiple seed entities. All-at-once retrievers tend to outperform path-based ones and KG agents, due to their higher recall of triples in ground-truth answer subgraphs. Such novel insights, that lead to significant leaderboard changes in the ranking of KG retrievers compared to established benchmarks, should guide the community in making more informed decisions when developing new solutions for KG retrieval. We show that the SynthKGQA-generated ground-truth subgraphs provide a stronger supervision signal when training KG retrievers compared to shortest paths, with downstream accuracy improvements of up to 30% for multi-hop questions. We also demonstrate that retrievers trained on the SynthKGQA-generated GTSQA dataset transfer well to human-curated KGQA datasets. These results, combined with the high flexibility of SynthKGQA, position it as a useful resource to train and evaluate reliable retrievers on any custom KG, contributing to the development of more trustworthy LLMs.

## Reproducibility Statement

The code to run all steps of the SynthKGQA pipeline (detailed in Section 3 and Appendix A) on a custom Knowledge Graph is made available in the supplementary material. We also include the full GTSQA dataset (train and test split) and the code to generate the question-specific graphs for the test questions (Appendix C.1; such graphs will be released together with the dataset). In the supplementary material we also provide all code to run the benchmarks and experiments (Sections 5 and 6) for the evaluated KG-RAG models, whose specifications are detailed in Appendix C.2.

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

## A  Additional Details on SynthKGQA

### A.1  Data Generation

We provide additional details on SynthKGQA's data generation pipeline, which is articulated in four consecutive steps (Figure 1).

**1. Seed subgraph sampling.** We first sample a connected subgraph $\mathcal{Q} \subset \mathcal{T}$ of the KG, which will provide the LLM with the context to generate a question, as asking the LLM to parse the entire KG in its input is unfeasible or impractical. Starting from a random entity $s \in \mathcal{E}$, the subgraph $\mathcal{Q}$ is obtained by progressively expanding a neighborhood around $s$ until the desired size (decided by the user, typically 20-100 edges) is reached. As we aim to construct multi-hop questions, it's important to ensure that the radius of $\mathcal{Q}$ grows quickly, even if the total number of edges in $\mathcal{Q}$ is kept contained. This means prioritizing depth over breadth in the expansion, while maintaining high-degrees of connectivity between nodes, to enable a variety of different isomorphism types for the answer subgraph. Our proposed sampling scheme for $\mathcal{Q}$, used in the construction of GTSQA, is shown in Algorithm 1; we denote by $d(x)$ the degree of a node $x$ (number of edges going in/out of it) and by $\mathcal{N}(x)$ its set of undirected neighbors.

**2. Datapoint generation.** The sampled subgraph $\mathcal{Q}$ is given to the LLM to generate one KGQA datapoint, which comprises of

- a question $q$ that can be answered with $k$ triples in $\mathcal{Q}$;

---

**Algorithm 1** Sampling scheme for the seed graph $\mathcal{Q}$

---

**Require:** KG $\mathcal{T} \subset \mathcal{E} \times \mathcal{R} \times \mathcal{E}$; starting node $s \in \mathcal{E}$; node limit $N_{\text{nodes}}$; edge limit $N_{\text{edges}}$

  $N \leftarrow \{s\}$
  $\mathcal{Q} \leftarrow \emptyset$
  **while** $|N| < N_{\text{nodes}}$ and $|\mathcal{Q}| < N_{\text{edges}}$ **do**
    $z \leftarrow \text{choice}(N, p = \text{Softmax}([d(x)^{-1} \text{ for } x \in N]))$          ▷ Sample node to expand
    $M \leftarrow \mathcal{N}(z) \setminus N$          ▷ Retrieve new neighbors of $z$
    **if** $|M| > 0$ **then**
      $n \leftarrow \text{choice}(M, p = \text{Softmax}([d(x)^{-1} \text{ for } x \in M]))$          ▷ Sample neighbor
      $\mathcal{Q} \leftarrow \mathcal{Q} \cup \{(h, r, t) \in \mathcal{T} \text{ s.t. } h = n, t \in N \text{ or } t = n, h \in N\}$    ▷ Add all connections to $n$
      $N \leftarrow N \cup \{n\}$
    **end if**
  **end while**

---

- the list of triples in $\mathcal{Q}$ required to reason over the question, i.e., the *ground-truth answer subgraph* $\mathcal{G} \subset \mathcal{Q}$;

- the answer to the question, which is required to be an entity $a \in \mathcal{E}$ appearing in $\mathcal{G}$;

- the list of entities explicitly mentioned in the question, i.e., the seed entities $\mathcal{S} \subset \mathcal{E}$, which also need to appear in $\mathcal{G}$;

- the SPARQL query $l_q$ which encodes the natural-language question $q$ in logical form.

In the prompt we specify the number $k$ of triples in $\mathcal{Q}$ that should be used to reason over the question (i.e., the number of triples in the ground-truth answer subgraph $\mathcal{G}$), and show examples in few-shot prompting style. The RDF identifiers in the knowledge base (e.g., QIDs and PIDs when working with Wikidata) are included in the prompt, together with labels, for all entities and relations appearing in $\mathcal{Q}$, in order to ensure consistency in the generation of the SPARQL query. Note that, while we use SPARQL as RDF query language since the Wikidata Query Service[2] is based on it, our pipeline can be adapted with minimal changes to use any other query language. An example (for $k = 2$) of the prompt used for the construction of GTSQA is shown below.

---

**LLM Prompt** ($k = 2$)

```
{ "role": "user", "content": "Based on the provided set of knowledge graph triples, please
generate a question that requires combining the information in exactly 2 of the provided
triples for answering. The answer should correspond to exactly one node in the provided graph,
be unique and not ambiguous. Make sure that all 2 of the selected triples are required for
answering the question and that they involve multiple different entities. Respond only with
```

- 'Question:' the generated question,
- 'Nodes mentioned in the question:' a semicolon-separated list of the nodes that are explicitly mentioned in the question,
- 'Answer:' the node corresponding to the correct answer with its QID,
- 'Triples used:' a semicolon-separated list of the triples used for answering the question,
- 'SPARQL query:' a SPARQL query to return all answers of the question from the Wikidata knowledge base.

```
"},
{ "role": "user", "content": "Graph: Quiaios (Q1024001)-country (P17)-Portugal (Q45);Savage
Nights (Q174785)-filming location (P915)-Portugal (Q45);Quiaios (Q1024001)-located in
the administrative territorial entity (P131)-Figueira da Foz (Q428459);Figueira da Foz
(Q428459)-country (P17)-Portugal (Q45);Savage Nights (Q174785)-filming location (P915)-Lisbon
(Q597);Portugal (Q45)-capital (P36)-Lisbon (Q597);Lisbon (Q597)-country (P17)-Portugal
```

---

[2]https://query.wikidata.org/

```
(Q45);Lisbon (Q597)-capital of (P1376)-Portugal (Q45);Francisco José Fernandes Costa
(Q357932)-country of citizenship (P27)-Portugal (Q45);Francisco José Fernandes Costa
(Q357932)-place of death (P20)-Figueira da Foz (Q428459);Francisco José Fernandes Costa
(Q357932)-position held (P39)-Minister of Foreign Affairs (Q4294919);Minister of Foreign
Affairs (Q4294919)-country (P17)-Portugal (Q45);Minister of Foreign Affairs (Q4294919)-applies
to jurisdiction (P1001)-Portugal (Q45);Francisco José Fernandes Costa (Q357932)-occupation
(P106)-politician (Q82955)
"},
{ "role":  "assistant", "content":  "
Question:  In which country did Francisco José Fernandes Costa die?,
Nodes mentioned in the question:  Francisco José Fernandes Costa (Q357932),
Answer:  Portugal (Q45),
Triples used:  Francisco José Fernandes Costa (Q357932)-place of death (P20)-Figueira da Foz
(Q428459); Figueira da Foz (Q428459)-country (P17)-Portugal (Q45),
SPARQL query:  SELECT ?answer WHERE {wd:Q357932 wdt:P20 ?place.  ?place wdt:P17 ?answer.}
"},
[...][**OMITTED - examples with different graph isomorphism types**]
{"role":  "user", "content":  "Graph:  {𝒬}"},
```

**3. Candidate validation.**   The query $l_q$ is executed against the KG to retrieve the full set of answers $\mathcal{A}$; the candidate datapoint is discarded if $\mathcal{A}$ does not contain the answer $a$ provided by the LLM. Similarly, it is discarded if any of the triples that the LLM considers necessary for answering the question, or any of the proposed seed entities, are not in the *full answer subgraph*, obtained by running the associated CONSTRUCT SPARQL query (i.e., the union of all triples in the KG realizing the query template, taking into account all valid substitutions of the variables in $l_q$; for more details, see the example in Appendix A.4).

**4. Augmentation and classification.**   We use a different LLM to paraphrase the question $q$ in more natural terms. This helps to mitigate any structure biases originating from the data generation pipeline and improves the naturalness of the question, as shown in the examples of questions and their paraphrased versions in Appendix A.4.

We classify the structure of $\mathcal{G}$ up to isomorphism and report this graph isomorphism type to enable a better evaluation based on question complexity and allow the user to perform additional filtering by discarding questions where the answer subgraph does not comply with desired requirements (e.g., disconnected graphs; graphs containing loops or hanging branches; graphs where the answer node is also used as a seed node, giving rise to self-answering questions). See Appendix A.2 for details.

While we check (during step 3) that all seed entities identified by the LLM are used in the SPARQL query, this does not guarantee that all of them are strictly necessary to answer the question, i.e., concretely impose additional constraints compared to the other seeds. Therefore, using the SPARQL query, we also determine whether the provided set of seed entities $\mathcal{S}$ is minimal or whether there is a subset $\mathcal{S}' \subsetneq \mathcal{S}$ of entities that suffices to answer the question (see Appendix A.3).

### A.2   Graph Isomorphisms

We use the notion of *graph isomorphism*, applied to the ground-truth answer subgraph $\mathcal{G}$, to provide a simple and objective measure of the complexity of a question in a KGQA dataset. This measure abstracts away the identity of the entities and relation types involved in the answer subgraph and only focuses on the number of seed entities, the number of hops separating them from the answer node and how the paths originating from each seed intersect. In the context of KGQA, this notion first appeared as *reasoning paths* (Das et al., 2022) and *semantic structures* (Li & Ji, 2022), and was then formulated in the same way that we will use it in Dutt et al. (2023) for the construction of GrailQA++.

We say that two questions have the same graph isomorphism type if their ground-truth answer subgraphs $\mathcal{G}$ are isomorphic as labeled graphs, when each node in $\mathcal{G}$ is labeled as "seed" (for seed nodes), "answer" (for the answer node) or "intermediate". This means that there exists a bijection of the sets of vertices of the two graphs that preserves both edges and labels. Note that, while KGs are directed graphs, when

Table 5: Isomorphism types of ground-truth answer subgraphs in GTSQA and their frequencies in the train and test split. In the graph visualisations, seed nodes are represented in red, while the answer node is marked in green. For the test set, we count separately questions where the answer subgraph is of an isomorphism type not present in the train set (unseen graph type, ugt), questions involving relation types not appearing in the train set (unseen relation type, urt) and questions where the answer subgraph is in-distribution with the train set (id). Percentages are based on the total sizes of the train and test set, respectively.

| Identifier | Visualisation | Dutt et al. (2023) | Train | | Test (ugt) | | Test (urt) | | Test (id) | |
|---|---|---|---|---|---|---|---|---|---|---|
| | | | count | % | count | % | count | % | count | % |
| (1) | | Iso-0 | 7658 | 25.1 | 0 | 0 | 100 | 6.17 | 50 | 3.08 |
| (2) | | Iso-1 | 8338 | 27.4 | 0 | 0 | 100 | 6.17 | 50 | 3.08 |
| (1)(1) | | Iso-2 | 4481 | 14.7 | 0 | 0 | 100 | 6.17 | 50 | 3.08 |
| (2)(1) | | Iso-3 | 1470 | 4.82 | 0 | 0 | 77 | 4.75 | 50 | 3.08 |
| ((1)(1)) | | Iso-4 | 0 | 0 | 125 | 7.71 | 0 | 0 | 0 | 0 |
| (3) | | Iso-5 | 1731 | 5.68 | 0 | 0 | 130 | 8.01 | 50 | 3.08 |
| (2)(2) | | Iso-6 | 0 | 0 | 139 | 8.57 | 0 | 0 | 0 | 0 |
| (3)(1) | | Iso-7 | 0 | 0 | 150 | 9.25 | 0 | 0 | 0 | 0 |
| ((2)(1)) | | Iso-8 | 0 | 0 | 65 | 4.01 | 0 | 0 | 0 | 0 |
| ((1)(1))(1) | | Iso-9 | 0 | 0 | 60 | 3.7 | 0 | 0 | 0 | 0 |
| (2(1)(1)) | | Iso-10 | 0 | 0 | 55 | 3.39 | 0 | 0 | 0 | 0 |
| (1)(1)(1) | | Iso-11 | 5526 | 18.1 | 0 | 0 | 48 | 2.96 | 50 | 3.08 |
| ((1)(1)(1)) | | Iso-12 | 144 | 0.47 | 0 | 0 | 0 | 0 | 0 | 0 |
| (2)(2)(1) | | Iso-13 | 24 | 0.08 | 0 | 0 | 0 | 0 | 0 | 0 |
| ((3)(1)) | | Iso-14 | 1 | 0 | 0 | 0 | 0 | 0 | 0 | 0 |
| (4)(1) | | Iso-15 | 15 | 0.05 | 0 | 0 | 0 | 0 | 0 | 0 |
| (4) | | Iso-16 | 0 | 0 | 50 | 3.08 | 0 | 0 | 0 | 0 |
| ((1)(1))(2) | | Iso-17 | 10 | 0.03 | 0 | 0 | 0 | 0 | 0 | 0 |
| (2)(1)(1) | | Iso-18 | 907 | 2.98 | 0 | 0 | 0 | 0 | 0 | 0 |
| ((2)(1)(1)) | | Iso-19 | 3 | 0.01 | 0 | 0 | 0 | 0 | 0 | 0 |
| ((1)(1))(1)(1) | | Iso-22 | 27 | 0.09 | 0 | 0 | 0 | 0 | 0 | 0 |
| ((2)(1))(1) | | Iso-23 | 5 | 0.02 | 0 | 0 | 0 | 0 | 0 | 0 |
| ((1)(1)(1))(1) | | Iso-25 | 18 | 0.06 | 0 | 0 | 0 | 0 | 0 | 0 |
| ((1)(1)(1))(2) | | Iso-26 | 1 | 0 | 0 | 0 | 0 | 0 | 0 | 0 |
| (1)(1)(1)(1) | | N/A | 0 | 0 | 123 | 7.58 | 0 | 0 | 0 | 0 |
| (1)(1)(1)(1)(1) | | N/A | 111 | 0.36 | 0 | 0 | 0 | 0 | 0 | 0 |
| (5) | | N/A | 7 | 0.02 | 0 | 0 | 0 | 0 | 0 | 0 |

computing isomorphism types we consider the answer subgraph as undirected, as KG retrievers should ideally be able to retrieve relevant edges independently of their direction. To provide identifiers for the different graph isomorphism types, the notation based on projections and intersections that is sometimes used in KG reasoning papers (e.g., in Das et al. (2022)) is unfortunately not able to describe complex graph structures without ambiguities. Since, as a data quality requirement, we only consider queries where the answer subgraph is a tree, we adopt a simplified/more readable version of the tree encoding scheme classically used in the AHU algorithm (Aho & Hopcroft, 1974). The answer subgraph is seen as a tree rooted in the answer node, while seed entities are the leaves of the tree. For each leaf, we write $(n)$ if $n$ is the distance of the leaf from its closest branching point (this represents a projection of length $n$ from the seed node). The intersection of paths from multiple seeds at a branching point is denoted by juxtaposition, e.g. $(n)(m)$. Further projections after an intersection point are represented using an additional level of brackets, e.g. $(k(n)(m))$, with $k$ the length of the projection (which can be omitted if $k = 1$). See Table 5 for notations and graphical representations of all the graph isomorphism types appearing in GTSQA.

## A.3 Minimality

A question $q$ is *non-minimal* if there exists a subset $\mathcal{S}' \subsetneq \mathcal{S}$ such that the SPARQL query $l_{q,\mathcal{S}'}$ corresponding to the subtree $\mathcal{G}_{\mathcal{S}'} \subset \mathcal{G}$ composed by the paths from the seeds in $\mathcal{S}'$ to the answer node $a$ returns the same set of answers as the original query $l_q$ (while, in general, we expect the set of answers of $l_{q,\mathcal{S}'}$ to be a strict superset of $\mathcal{A}$). In the case of non-minimal questions, we identify the smallest such $\mathcal{S}'$ (there can be more than one) and classify the graph isomorphism type of $\mathcal{G}_{\mathcal{S}'}$, which should be considered as the minimal subgraph(s) sufficient to answer the question. Note that non-minimal questions are still perfectly valid, as all seed entities are mentioned in the question and the information they imply should be therefore retrieved from the KG; however, they present "shortcuts" to the answer that effectively reduce the question complexity, compared to what is measured by the graph isomorphism type of the ground-truth answer subgraph.

For example, the question "Which musical instrument is played by both Yehonatan Geffen's child and Francis Lickerish?" has two seed entities: $\mathcal{S} = \{$Yehonatan Geffen (Q2911403), Francis Lickerish (Q3720616)$\}$. The corresponding SPARQL query

```
SELECT ?answer WHERE
{ wd:Q2911403 wdt:P40 ?child . ?child wdt:P1303 ?answer .
wd:Q3720616 wdt:P1303 ?answer . }
```

has a unique answer in Wikidata, namely *guitar*. The ground-truth answer subgraph contains three edges, (Yehonatan Geffen; child; Aviv Geffen), (Aviv Geffen; instrument; guitar), (Francis Lickerish; instrument; guitar), with isomorphism type (2)(1). The SPARQL query decomposes as intersection of two projections, originating from the seed entities. We can look at them separately.

- For seed node *Yehonatan Geffen*, the 2-hop path connecting it to the answer node is encoded by the query

  ```
  SELECT ?answer WHERE
  { wd:Q2911403 wdt:P40 ?child . ?child wdt:P1303 ?answer . }
  ```

  which returns three different answers: *guitar, piano, voice*.

- For seed node *Francis Lickerish*, the 1-hop path connecting it to the answer node is encoded by the query

  ```
  SELECT ?answer WHERE
  {wd:Q3720616 wdt:P1303 ?answer.}
  ```

  which returns a single answer, *guitar*.

Therefore, this question is non-minimal, as it can be satisfyingly answered using just the seed entity *Francis Lickerish* (while using the other seed entity alone is not sufficient). The minimal answer subgraph is {(Francis Lickerish; instrument; guitar)}, with isomorphism type (1).

### A.4 Examples

We display, for exemplificative purposes, one datapoint from the train set of GTSQA, generated from Wikidata with the SynthKGQA framework.

---

**Example**

```
"id": 40513,

"question": "Who directed the Italian film, originally in French, that is based on 'The
Vicomte of Bragelonne: Ten Years Later'?",

"paraphrased_question": "Who was the director of the Italian film, originally in French,
inspired by 'The Vicomte of Bragelonne: Ten Years Later'?",

"seed_entities": ["Italy (Q38)", "French (Q150)", "The Vicomte of Bragelonne: Ten Years
Later (Q769001)"],

"answer_node": "Fernando Cerchio (Q503508)",

"answer_subgraph": [["Le Vicomte de Bragelonne (Q3228085)", "country of origin (P495)",
"Italy (Q38)"], ["Le Vicomte de Bragelonne (Q3228085)", "original language of film or TV show
(P364)", "French (Q150)"], ["Le Vicomte de Bragelonne (Q3228085)", "based on (P144)", "The
Vicomte of Bragelonne: Ten Years Later (Q769001)"], ["Le Vicomte de Bragelonne (Q3228085)",
"director (P57)", "Fernando Cerchio (Q503508)"]],

"sparql_query": "SELECT ?answer WHERE { ?film wdt:P495 wd:Q38; wdt:P364 wd:Q150; wdt:P144
wd:Q769001; wdt:P57 ?answer.}",

"all_answers_wikidata": ["Q503508", "Q679016"],

"full_answer_subgraph_wikidata": [["Q2260875", "P495", "Q38"], ["Q2260875", "P364",
"Q150"], ["Q2260875", "P144", "Q769001"], ["Q226087", "P57", "Q679016"], ["Q322808", "P495",
"Q38"], ["Q3228085", "P364", "Q150"], ["Q3228085", "P144", "Q769001"], ["Q3228085", "P57",
"Q503508"]],

"all_answers_wikikg2": ["Q503508"],

"full_answer_subgraph_wikikg2": [["Q3228085", "P364", "Q150"], ["Q3228085", "P57",
"Q503508"], ["Q3228085", "P144", "Q769001"], ["Q3228085", "P495", "Q38"]],

"n_hops": 2,

"graph_isomorphism": "((1)(1)(1))",

"redundant": True,

"minimal_graph_isomorphism": "((1)(1))",

"minimal_seeds_and_queries": {"Q150-Q769001": "SELECT ?answer WHERE { ?a wdt:P364 wd:Q150.
?a wdt:P57 ?answer. ?a wdt:P144 wd:Q769001.}"}

"test_type": [],
```

---

Note that `answer_node` and `answer_subgraph` are, respectively, the answer node $a \in \mathcal{E}$ and ground-truth answer subgraph $\mathcal{G} \subset \mathcal{T}$ generated by the LLM together with the question. The `sparql_query` is then executed on Wikidata and WikiKG2 to retrieve all answers in the KGs (`all_answers_wikidata`; `all_answers_wikikg2`) and, after converting it to CONSTRUCT form, the full answer subgraphs realizing the query (`full_answer_subgraph_wikidata`; `full_answer_subgraph_wikikg2`). For the example above, we find that Wikidata (but not WikiKG2) contains one more acceptable answer (Henri Decoin, Q679016), due to the existence in the KG of a second movie satisfying all requirements in the question (Le Masque de fer, Q2260875); as a consequence, `full_answer_subgraph_wikidata` contains four more edges compared to $\mathcal{G}$, arising from these extra valid substitutions for the `?film` and `?answer` variables. Note that we only consider the answer graph $\mathcal{G}$ when computing the `graph_isomorphism`, as that encodes the logical steps required to reason over the question. However, the full answer subgraph should be used as target to evaluate the performance of KG retrievers.

`n_hops` measures the maximum distance (in $\mathcal{G}$) between a seed entity and the `answer_node`; it is determined in a unique way by `graph_isomorphism`. The question in the example contains redundant information

(`redundant`); in the `minimal_seeds_and_queries` dictionary we provide the minimal set(s) $\mathcal{S}'$ of seed entities and the respective SPARQL queries $l_{q,\mathcal{S}'}$, as explained in Appendix A.3. The graph isomorphism of the minimal $\mathcal{G}_{\mathcal{S}'}$ can be found in `minimal_graph_isomorphism`. Finally, the attribute `test_type` is only used for questions in the test split of GTSQA, to classify their generalization type (in-distribution, unseen graph type, unseen relation type; see Section 4).

To judge the naturalness and reasonability of questions produced by the SynthKGQA framework, we present three randomly sampled questions for every graph isomorphism type in GTSQA with at least 50 questions in the training set. We observe that the paraphrasing operated by the LLM helps to make the questions sound more natural and human-like.

---

**Example questions and their paraphrased versions**

   Graph isomorphism type (1)

     Question: On which continent is Palmer Land located?
     Paraphrased Question: Which continent is Palmer Land situated on?

     Question: What was the military rank of Pierre Gaston-Mayer?
     Paraphrased Question: What military rank did Pierre Gaston-Mayer hold?

     Question: What was the noble title held by Sir Edward Kerrison, 1st Baronet?
     Paraphrased Question: What noble title did Sir Edward Kerrison, 1st Baronet, hold?

   Graph isomorphism type (2)

     Question: Who is the head of government of the capital of the canton of Mérignac-1?
     Paraphrased Question: Who is the leader of the government in the capital of the Mérignac-1 canton?

     Question: Who is the producer of the series that the episode 1930-talet is part of?
     Paraphrased Question: Who produced the series that includes the episode titled "1930-talet"?

     Question: Who composed the anthem of Tuvalu?
     Paraphrased Question: Who is the composer of Tuvalu's national anthem?

   Graph isomorphism type (3)

     Question: In which country is the administrative entity in which Arhavi is located, located?
     Paraphrased Question: Which country is home to the administrative region where Arhavi is situated?

     Question: Where is the mother of the spouse of Ruprecht V of Nassau buried?
     Paraphrased Question: Where is Ruprecht V of Nassau's mother-in-law buried?

     Question: Who is the architect of the building occupied by the sports team that Raitis Grafs played for?
     Paraphrased Question: Who designed the building where the sports team that Raitis Grafs played for is located?

   Graph isomorphism type (1)(1)

     Question: Which company founded by Henry Herbert Collier has its headquarters in Plumstead?
     Paraphrased Question: Which company, established by Henry Herbert Collier, is headquartered in Plumstead?

     Question: Which resident of District 2 was killed by Thresh?
     Paraphrased Question: Who was the resident of District 2 that Thresh killed?

     Question: Which film produced by UK Film Council was based on The Picture of Dorian Gray?
     Paraphrased Question: What movie produced by the UK Film Council was inspired by The Picture of Dorian Gray?

   Graph isomorphism type (1)(1)(1)

---

Question: Which singer, born in Suphan Buri, holds citizenship of Thailand and plays the guitar?
Paraphrased Question: Which Thai singer from Suphan Buri plays the guitar?

Question: Which film produced by Metro-Goldwyn-Mayer, whose main subject is baseball, was produced by Clarence Brown?
Paraphrased Question: What baseball-themed film made by Metro-Goldwyn-Mayer was directed by Clarence Brown?

Question: Who participated in the 2002 FIFA World Cup and played for both GNK Dinamo Zagreb and Sevilla FC?
Paraphrased Question: Which player took part in the 2002 FIFA World Cup and played for both GNK Dinamo Zagreb and Sevilla FC?

Graph isomorphism type (2)(1)

Question: Which team that plays association football is the occupant of the stadium operated by Fenerbahçe Sports Club?
Paraphrased Question: Which football team plays at the stadium managed by Fenerbahçe Sports Club?

Question: What is the city that is both the headquarters location of the developer of Star Wars: Shadows of the Empire (video game) and the narrative location of Sudden Impact?
Paraphrased Question: Which city serves as the headquarters of the developer for the video game Star Wars: Shadows of the Empire and is also the setting for Sudden Impact?

Question: Which street, used for utility cycling, serves as a terminus for a street that is itself a terminus at Frankfurter Tor?
Paraphrased Question: Which street designed for utility cycling acts as a terminus for another street that ends at Frankfurter Tor?

Graph isomorphism type ((1)(1)(1))

Question: Which cast member acted in a Spanish-language film that was filmed in Mexico and belongs to the exploitation film genre?
Paraphrased Question: Which cast member starred in a Spanish-language film shot in Mexico that falls under the exploitation genre?

Question: Who is the head of government of the municipality that is contained within Valdizarbe and shares borders with both Uterga and Enériz?
Paraphrased Question: Who is the head of government for the municipality in Valdizarbe that borders both Uterga and Enériz?

Question: Which award was received by the English-language film directed by Terence Davies and starring Freda Dowie?
Paraphrased Question: What award did the English-language film directed by Terence Davies and featuring Freda Dowie win?

Graph isomorphism type (2)(1)(1)

Question: Which goalkeeper who participated in the 1958 FIFA World Cup died in a town located in the Eastern Time Zone?
Paraphrased Question: Which goalkeeper from the 1958 FIFA World Cup passed away in a town in the Eastern Time Zone?

Question: Which businessperson who was a cast member of a reality television show also attended the University of Pennsylvania?
Paraphrased Question: Which reality TV star who is also a businessperson went to the University of Pennsylvania?

Question: Which person died of pneumonia in Vienna and has an asteroid belt minor planet named after them?
Paraphrased Question: Who passed away from pneumonia in Vienna and has a minor planet in the asteroid belt named in their honor?

Graph isomorphism type (1)(1)(1)(1)(1)

```
     Question:  Which musical film directed by Challis Sanderson features both Robb Wilton and
     Kitty McShane as cast members and has Desmond Dickinson as its director of photography?
     Paraphrased Question:  What musical film directed by Challis Sanderson stars Robb Wilton
     and Kitty McShane, with Desmond Dickinson as the director of photography?

     Question:  Which video game is both an action and platform game, distributed on Nintendo
     game card for Nintendo DS and published in North America?
     Paraphrased Question:  What action and platform game is available on a Nintendo game card
     for the Nintendo DS and was published in North America?

     Question:  Which minor planet in the asteroid belt was discovered at Osservatorio
     Astronomico Sormano by both Francesco Manca and Piero Sicoli and is directly preceded by
     9110 Choukai?
     Paraphrased Question:  What is the name of the minor planet in the asteroid belt,
     discovered by Francesco Manca and Piero Sicoli at the Osservatorio Astronomico Sormano,
     that comes right before 9110 Choukai?
```

## A.5  Aggregation Operators

In the construction of GTSQA we focus on entity-valued conjunctive queries, since these provide the cleanest signal for evaluating KG retrieval abilities (as aggregation logic can be typically delegated to the LLM performing the final reasoning, once all the relevant paths have been retrieved). However, with small modifications to the prompt, SynthKGQA can also generate queries that go beyond simple path-retrieval, by requesting the SPARQL query to include the desired aggregation operators. See example prompt below, using the COUNT operator. Notice that, since the subgraph $\mathcal{Q}$ that is shown to the LLM to generate the question might not contain all the edges required to perform aggregation (e.g., all the nodes that need to be counted by COUNT), the final answer needs to be generated after submitting the SPARQL query and retrieving the full answer subgraph.

---

**LLM Prompt for aggregation queries**

```
{ "role":  "user", "content":  "Based on the provided set of knowledge graph triples, please
generate a question that involves counting.  The question should be grounded in one or more
edges in the provided graph, but the graph is not guaranteed to contain all answers.  Don't
worry if the showed graph only contains a subset of what you believe are all the answers -
this will be fixed when we submit the SPARQL query to Wikidata, to retrieve all correct answers.
Apply your internal knowledge and common sense to ensure that you respond with a question that
is not ambiguous.  Respond only with
```

- 'Question:'  the generated question,
- 'Nodes mentioned in the question:'  a semicolon-separated list of the nodes that are explicitly mentioned in the question,
- 'Answer:'  one of the correct items to be counted appearing in the graph, with its QIDs,
- 'Triples used:'  a semicolon-separated list of the triples used for answering the question,
- 'SPARQL query:'  a SPARQL query, using the COUNT operator, to return all answers of the question from the WikiData knowledge base.

```
"},
{ "role":  "user", "content":  "Graph:  Richard Laqueur (Q100246)-occupation (P106)-historian
(Q201788);Peter Schreiner (Q98564)-occupation (P106)-historian (Q201788);Richard Laqueur
(Q100246)-educated at (P69)-University of Bonn (Q152171);Sabine Mainberger (Q1589597)-employer
(P108)-University of Bonn (Q152171);Richard Laqueur (Q100246)-languages spoken, written or
signed (P1412)-German (Q188);Peter Schreiner (Q98564)-member of (P463)-Göttingen Academy
of Sciences (Q414188);Richard Laqueur (Q100246)-country of citizenship (P27)-Germany
(Q183);Peter Schreiner (Q98564)-country of citizenship (P27)-Germany (Q183);University of
Bonn (Q152171)-country (P17)-Germany (Q183);Germany (Q183)-official language (P37)-German
(Q188);Sabine Mainberger (Q1589597)-country of citizenship (P27)-Germany (Q183);Göttingen
Academy of Sciences (Q414188)-country (P17)-Germany (Q183);Richard Laqueur (Q100246)-conflict
(P607)-World War I (Q361);William Otway Boger (Q8016402)-conflict (P607)-World War I (Q361)
```

```
"},
{ "role":  "assistant", "content":  "
Question:  What is the number of countries where Richard Laqueur spent time studying?,
Nodes mentioned in the question:  Richard Laqueur (Q100246),
Answer:  Germany (Q183),
Triples used:  Richard Laqueur (Q100246)-educated at (P69)-University of Bonn (Q152171);
University of Bonn (Q152171)-country (P17)-Germany (Q183),
SPARQL query:  SELECT (COUNT(DISTINCT ?answer) AS ?count) WHERE  wd:Q100246 wdt:P69 ?university .
?university wdt:P17 ?answer
"},
[...][**OMITTED - examples with different graph isomorphism types**]
{"role":  "user", "content":  "Graph:  {Q}"},
```

Below, we show examples generated by SynthKGQA on Wikidata involving aggregation and negation.

---

**Example: COUNT**

"paraphrased question":  "How many songs by Neal McCoy were released under Atlantic Records?",

"seed nodes":  ["Neal McCoy (Q1973050)", "Atlantic Records (Q202440)"],

"sparql query":  "SELECT (COUNT(DISTINCT ?answer) AS ?count) WHERE  ?answer wdt:P175 wd:Q1973050; wdt:P264 wd:Q202440 .  ",

"answer":  21,

"all answers wikidata":  ["Q17013036", '"Q17039085", ...]

---

**Example: FILTER NOT EXISTS**

"paraphrased question":  "Who are the mushers that took part in the 1997 Yukon Quest without winning it?",

"seed nodes":  ["musher (Q500097)", "1997 Yukon Quest (Q18579418)"],

"sparql query":  "SELECT ?answer WHERE  ?answer wdt:P106 wd:Q500097 .  ?answer wdt:P1344 wd:Q18579418 .  FILTER NOT EXISTS  ?answer wdt:P1346 wd:Q18579418 ",

"all answers wikidata":  ["Q1375042", "Q18580343", ...]

---

# B  Statistics of GTSQA

We use ogbl-wikikg2 (Hu et al., 2020) as the base KG to construct GTSQA. This is a KG extracted from a 2015 Wikidata dump, containing a curated set of 2.5M nodes and 535 relation types, that we find are good candidates for the construction of natural-sounding questions. As part of the validation and filtering pipeline, by leveraging the generated SPARQL queries, we reject datapoints where any of the edges in the ground-truth answer subgraph inside ogbl-wikikg2 encode stale facts, i.e., edges that are not contained in the most up to date version of Wikidata[3]. In the future, GTSQA can be easily kept up to date by repeating the filtering process against new Wikidata dumps, or by replacing stale ground-truth subgraphs with the current ones, as retrieved from Wikidata via the provided SPARQL queries.

An overall view of the statistics of GTSQA is provided in Table 1. The questions in the dataset involve 68,520 unique entities, drawn from ogbl-wikikg2. The distribution of relation types is shown in more detail in Figure 5: out of the 368 unique relation types used in the dataset, only the top-200 (when sorting by overall frequency) appear in the train set, while the remaining ones are reserved for testing. Figure 6 shows the distribution of the number of seed entities, hops (in the ground-truth answer subgraph) and answers. Questions in the test set are significantly harder, requiring to perform more hops, or to combine reasoning chains from multiple seed entities. Our data-filtering pipeline focuses on selecting highly-factual and non-ambiguous questions; as a consequence, 73.9% and 85.9% of questions in the train and test set, respectively, have a single answer in Wikidata, and only a negligible fraction have more than five.

---

[3]At the time of dataset construction: https://dumps.wikimedia.org/wikidatawiki/20250720/

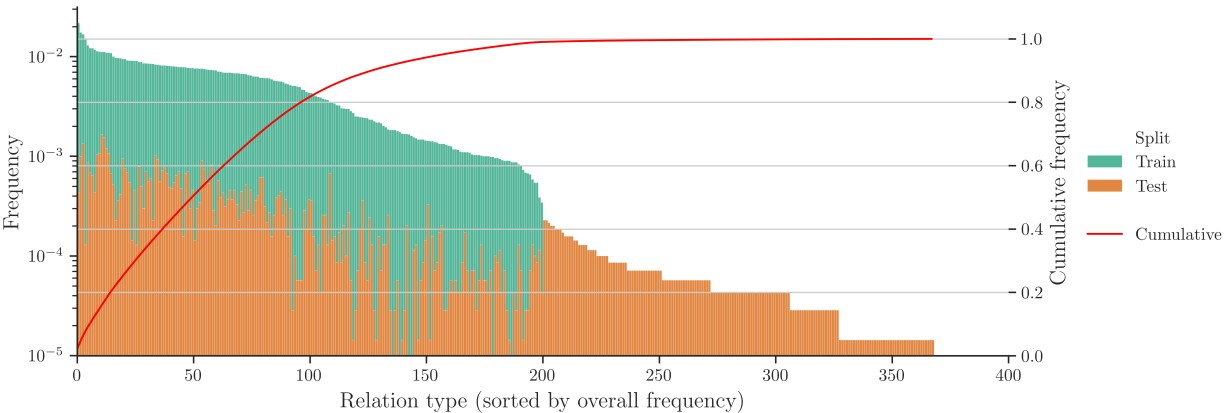

Figure 5: Frequency of relation types of edges in the ground-truth answer subgraphs of questions in GTSQA. The 168 least-occurring relation types (tail of the distribution) are reserved for questions in the test set, to test zero-shot generalization abilities of KG retriever models.

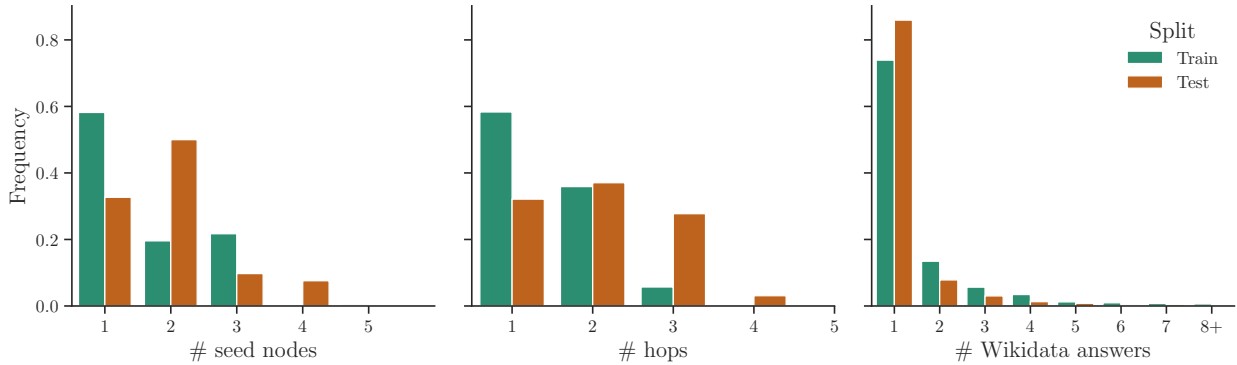

Figure 6: Statistics on the number of seed entities (*left*), the number of answer entities in Wikidata (*right*), and the maximum number of hops from seed to answer nodes along ground-truth paths (*center*), for questions in the train and test set of GTSQA.

We also compare the size of the ground-truth answer subgraph and the full answer subgraph retrieved from Wikidata by submitting the SPARQL query in CONSTRUCT form. As recalled in Appendix A.4, the full subgraph may contain additional edges, originating from the presence of multiple answers and/or multiple choices for the intermediate entities not specified in the query. In practice, as shown in Figure 7, for GTSQA this excess in the number of edges remains always limited (only in 28.4% and 15.5% of train and test questions, respectively, the two graphs do not coincide). Finally, we report on statistics on non-minimal questions in the train set of GTSQA (as stressed in Section 4, no redundancy is present in test questions). We find that 25.76% of training questions are non-minimal; Figure 8 shows the distributions of graph isomorphism types for the ground-truth answer subgraph $\mathcal{G}$ and the minimal subgraph $\mathcal{G}_{\mathcal{S}'}$ (see Appendix A.3).

## C   Additional Details on Experiments

### C.1   Construction of Question-Specific Graphs

An often overlooked fact in KGQA benchmarks is that many state-of-the-art KG retrievers, especially those that need to perform breadth/depth-first exploration of the KG from the seed entities (e.g., Luo et al.

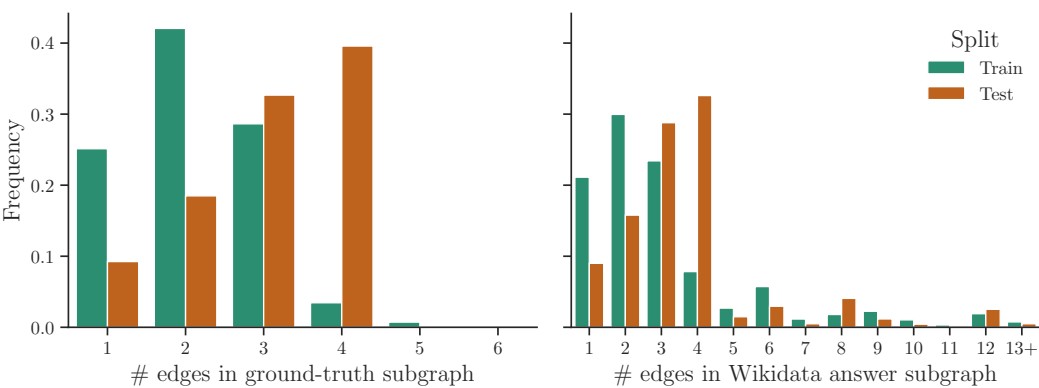

Figure 7: Statistics on the number of edges in the ground-truth answer subgraph (*left*) and the full answer subgraph in Wikidata (*right*), for questions in the train and test set of GTSQA. The full answer subgraph can contain more edges than the ground-truth subgraph, if the question has multiple answers, or if any of the intermediate nodes is not uniquely determined.

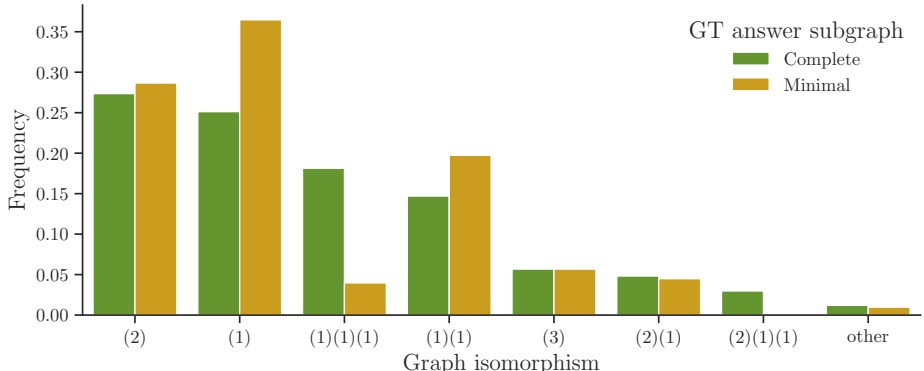

Figure 8: Frequency of main graph isomorphism types in the train set of GTSQA, distinguishing between the complete ground-truth answer subgraph $\mathcal{G}$ and the minimal subgraph $\mathcal{G}_{\mathcal{S}'}$.

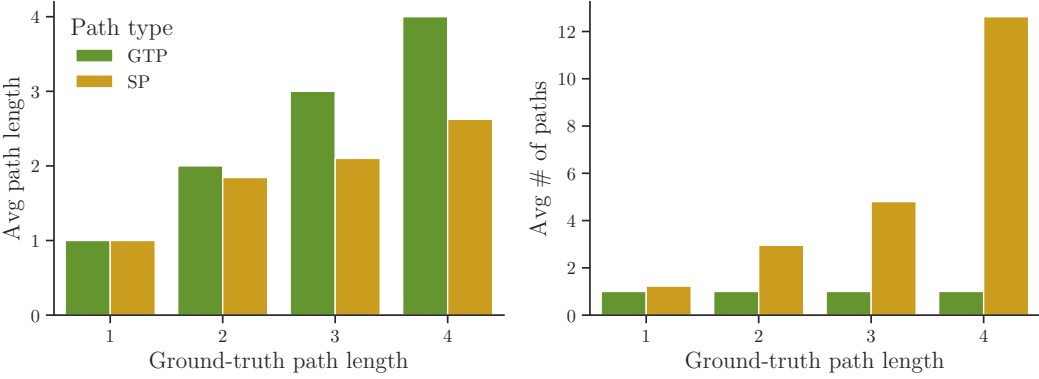

Figure 9: Comparison of ground-truth paths (GTP) and shortest paths (SP) connecting a question seed node to the answer node, in the train split of GTSQA. *Left:* for questions where the ground-truth paths require multiple hops, the distance from the seed node to the answer node along the shortest path increases sub-linearly (shortcuts). *Right:* as the distance between seed and answer node increases, the number of parallel paths of minimal length between them grows exponentially.

Table 6: Overlap of triples in ground-truth (GT) answer subgraph and triples on shortest paths (SP) between seed nodes and answer nodes. Questions are grouped by the maximum number of hops in the ground-truth subgraph (# hops); the other columns report the average metric values.

| # hops | % GT triples in SP | % SP triples in GT | # GT triples | # SP triples |
|--------|--------------------|--------------------|--------------|--------------|
| 1 | 100.0 | 91.0 | 2.37 | 2.67 |
| 2 | 89.8 | 52.7 | 3.08 | 19.6 |
| 3 | 54.7 | 29.6 | 3.6 | 46.3 |
| 4 | 27.5 | 13.3 | 4 | 100.4 |

(2024), Luo et al. (2025)) or that include Graph Neural Networks in their architectures (e.g., Mavromatis & Karypis (2025), Li et al. (2025)), are too expensive to run on KGs that have more than $\sim 10^5$ edges. This limitation, in practice, is tackled by performing the retrieval on a smaller subgraph of the KG, which is independently sampled for each question, e.g., by taking the full $k$-hop neighborhood of the seed entities in the KG, and then pruning it down to a few tens of thousands of edges with algorithms like Personalized PageRank (Page et al., 1998). This was the case for Luo et al. (2024), where such question-specific graphs were constructed (with $k = 2$) from Freebase (Bollacker et al., 2008) for the questions in WebQSP (Yih et al., 2016) and CWQ (Talmor & Berant, 2018). Note that, although questions in CWQ can require up to four hops, the corresponding question-specific graphs were still only constructed from 2-hop neighborhoods of seed entities. The graphs were then used by many others in following papers for benchmarking new models on these two widely-used datasets. They are not, however, part of the official datasets, hence there is no guarantee on their adoption. It is important to note, in fact, that the selection of these starting graphs can strongly impact the final performance statistics, potentially over-representing (if they are too easy/small) or under-representing (if they are not checked to still contain ground-truth paths) the retriever's capabilities. For this reason, retrievers that are benchmarked on different sets of questions-specific graphs should not be directly compared (even though the test questions, and the underlying full KG, are the same), and care should be taken when comparing them with models that instead are able to perform retrieval from the full KG (e.g., Sun et al. (2024)). However, this crucial detail on experimental setup is often not reported in papers, making comparisons unreliable.

To address this problem, together with GTSQA we release an official set of question-specific graphs (each containing up to 30,000 edges) for all questions in the train and test set. These are the only graphs that should be used by anyone wishing to train or benchmark KG-RAG models on GTSQA when retrieving from the full KG is not possible, to ensure fair comparisons. They are extracted from ogbl-wikikg2 with a similar approach to Luo et al. (2024), starting from the full undirected 3-hop (4-hop for questions requiring 4 hops) neighborhood of seed entities, and then pruning it down to the edges connecting the nodes with the top-2500 scores as assigned by Personalized PageRank (with personalization values concentrated in the seed nodes). If any of the edges in the (full) ground-truth answer subgraph of the question have been dropped as a result of pruning, they are re-added to the graph to guarantee that perfect retrieval is still possible. However, we observe empirically that this final step can unfairly bias retrievers towards the ground-truth edges that have been re-added. To ensure a challenging task, we also add to the graph (as confounders) all edges along paths originating from the seed nodes, of the same metapath (sequence of relation types) as the corresponding ground-truth paths that lead to the answer node.

## C.2 Specifications of Evaluated Models

We provide details and specifications on the KG-RAG models included in our benchmarks. For all of them, we follow original implementations as close as possible.

**Think-on-Graph (Sun et al., 2024), Plan-on-Graph (Chen et al., 2024)** We modified the original codebases (with the PoG one being based on the one from ToG) to perform retrieval from the ogbl-wikikg2 KG. The search and prune steps of the KG exploration algorithm use a width of 5 and a maximum depth of 4 to enable the retrieval of all paths in the ground-truth answer subgraphs of multi-seed, multi-hop questions

in the test set of GTSQA. Note that ToG, by its original implementation, reverts to answering using only the LLM knowledge if the maximum search depth is reached without the model being confident it has retrieved enough information; in this case, we treat the retrieved subgraph as being empty. The LLM performing the graph exploration (and task decomposition and memory updating for PoG) is the same used for the final reasoning, namely GPT-5-mini.

**SR (Zhang et al., 2022)**  As in the original implementation, we use RoBERTa$_{\text{BASE}}$ (Liu et al., 2019) to predict the next relation type $r_N \in \mathcal{R} \cup \{\text{END}\}$ in a path $(r_1, \dots, r_{N-1})$ from seed to answer node, conditioning on the question $q$. This is performed by fine-tuning the text-encoder to align the embeddings of $[q; r_1; \dots; r_{N-1}]$ and $r_N$, with a contrastive approach that uses positive and negative pairs constructed from ground-truth paths (from seed to answer) in the train set. At inference time, paths are predicted by conducting a beam search based on possible continuation candidates; we impose a maximum path length of 4 and set the number of beams to 5. The subgraph is retrieved by looking for all possible (undirected) realizations of the predicted relation paths in the KG starting from the seed entities. Note that, while the original implementation used a Neural State Machine (He et al., 2021) to perform the final reasoning on the retrieved subgraph, we instead use an LLM, to align the last step of the pipeline with the other KG-RAG models evaluated in the paper.

**Reasoning on Graphs (Luo et al., 2024)**  As in the original paper, we fine-tune LLama-2-Chat-7B (Touvron et al., 2023) to auto-regressively predict the relation paths $(r_1, \dots, r_N)$ (and specifying the direction of each edge), originating from the seed nodes, that should be useful to answer a question $q$. We use all ground-truth paths for the questions in the train set as fine-tuning data. At inference time we ask the LLM to propose relation paths using beam search (5 beams) which are then used to retrieve the subgraph via breadth-first search. We adopt the plug-and-play version of RoG, which allows us to use a different LLM (GPT-5-mini) to perform the final reasoning on the retrieved triples.

**Graph-Constrained Reasoning (Luo et al., 2025)**  While RoG can hallucinate non-existing relation paths, the follow-up work GCR constraints the path decoding to actual paths in the KG. However, this comes at significant costs in terms of overhead, as it requires to first index in a KG-Trie all paths (up to a fixed length) originating from the seed entities, retrieved via depth-first traversal of the graph. We find that, even when working with the smaller question-specific graphs from Appendix C.1, building such index for paths of length $> 2$ requires an impractical amount of time, which strongly limits the applicability of the method to real-world scenarios where the index needs to be built on-the-fly (as it happens for questions that have not been seen before). Thus, we test GCR with a maximum path length of 2 (which is also the default for experiments in Luo et al. (2025)), despite being aware that a significant fraction of questions in GTSQA require reasoning over longer paths. While we still include GCR in the benchmark of GTSQA, these limitations lead us to exclude GCR from the case-study in Section 6. As in the original implementation, we use LLama-3.1-8B-Instruct (AI@Meta, 2024) as LLM to generate paths, and fine-tune it with the same data used for RoG. At inference time, we generate 10 paths from the seed entities through graph-constraint decoding, and then discard duplicated paths to obtain the final retrieved subgraph.

**SubgraphRAG (Li et al., 2025)**  SubgraphRAG assigns relevance scores to all edges in the question-specific graphs (Appendix C.1), by combining text embeddings with message passing. In particular, $p((h, r, t)|q) \propto \text{MLP}([z_q; z_h; z_r; z_t; z_\tau])$, where $z_q, z_h, z_r, z_t$ are text embeddings from gte-large-en-v1.5 (Li et al., 2023) for the question $q$ and the labels of $h, r, t$. The embedding $z_\tau$ is constructed from the GNN embeddings of the $h$ and $t$ nodes, after 2 layers of message passing starting from the one-hot representation of nodes in the graph provided by the labeling trick (**1** if the node is a seed entity, **0** otherwise; Zhang et al. (2021)). We train GNN and MLP with cross-entropy loss to assign high scores to the triples in the ground-truth answer subgraphs, on the train split of GTSQA. At inference time, we retrieve the subgraph consisting of the triples with top-200 scores, as in the implementation of Li et al. (2025).

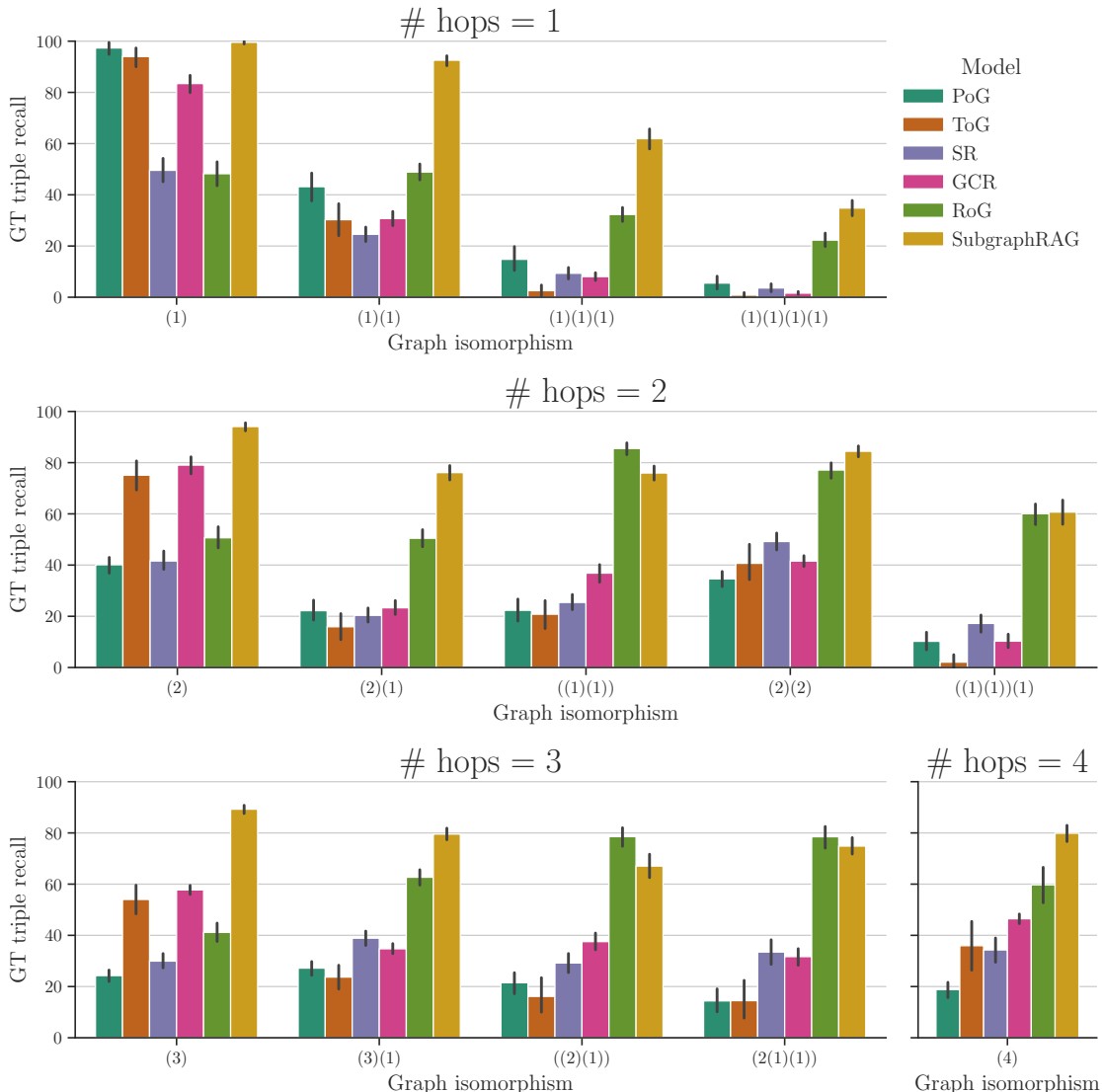

Figure 10: Recall of triples in ground-truth answer subgraph, for KG-RAG models on different graph isomorphism types.

## C.3  Additional Results

Figure 10 presents a comparison of the recall of ground-truth triples for the evaluated KG-RAG models on GTSQA, complementing the data in Figure 2. For trainable models, we also show the results disaggregated by generalization type of the test question in Figure 11.

In Table 7 we provide a comparison of the relative ranking of the evaluated KG retrievers across three different KGQA datasets: WebQSP (Yih et al., 2016), CWQ (Talmor & Berant, 2018) and GTSQA. Evaluation is performed on question-specific graphs, constructed in (Luo et al., 2024) for WebQSP and CWQ and – with a compatible pipeline – in Appendix C.1 for GTSQA. We see significant differences in the relative rankings of models across the datasets. We attribute this in part to the low factual correctness of WebQSP and CWQ (reported at around 50% in Zhang et al. (2025)) and to the issues in the construction of the question-specific graphs for CWQ highlighted in Appendix C.1. All this makes it intrinsically difficult to judge the value of benchmarks on WebQSP and CWQ.

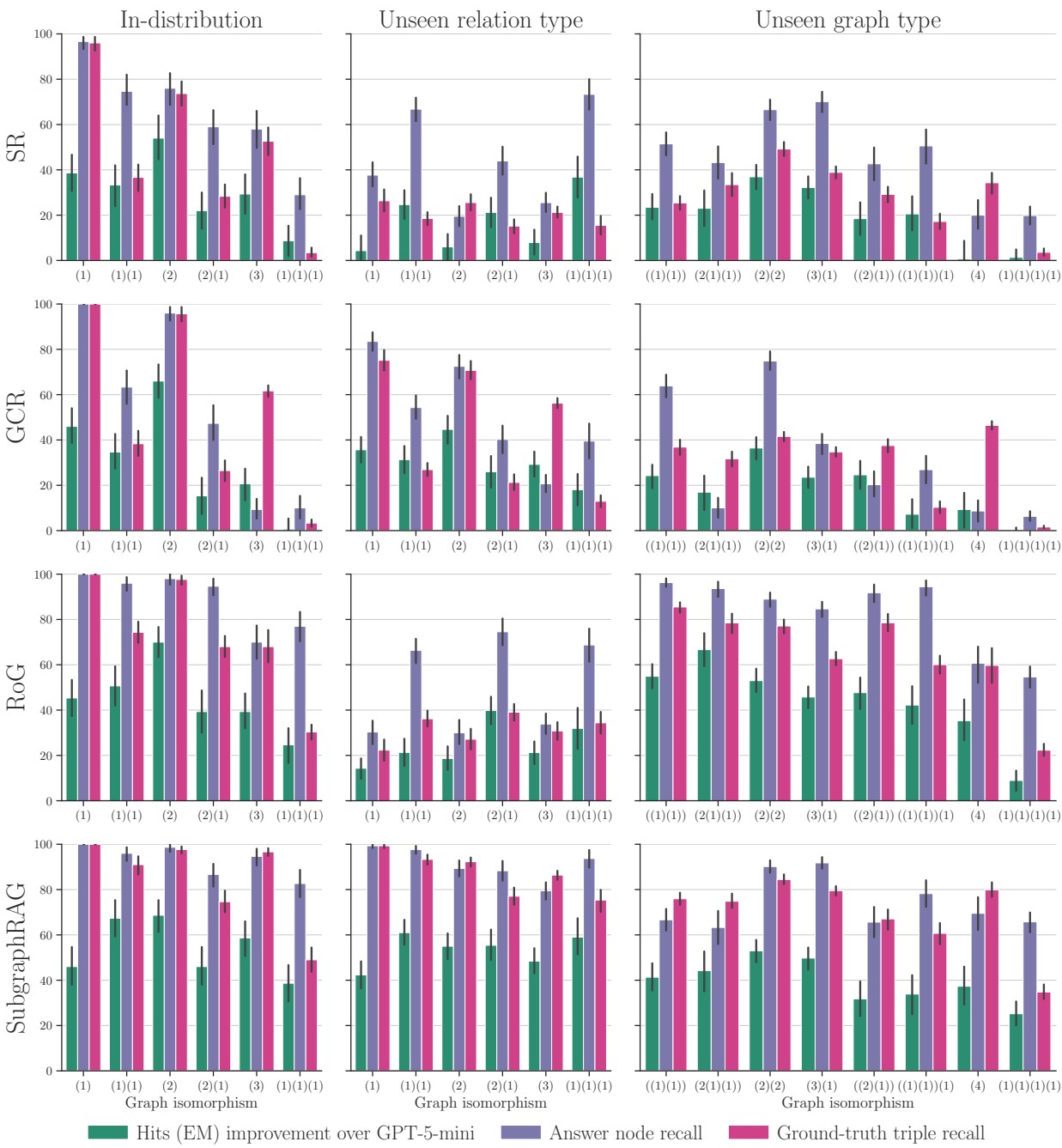

Figure 11: Detailed analysis of generalization abilities of trainable KG-RAG models, on different graph isomorphism types. We measure EM performance in terms of difference with the EM of the baseline (GPT-5-mini, no RAG).

Table 7: Hits (EM) scores of KG retrievers on WebQSP, CWQ and GTSQA. Bold and underline denote the best and second best model. All retrievers use GPT-4o-mini as final reasoning LLM. The results for WebQSP and CWQ are taken from Li et al. (2025), Luo et al. (2025) and Chen et al. (2024).

| Model | WebQSP | CWQ | GTSQA |
|---|---|---|---|
| PoG | 87.3 | 75.0 | 32.9 |
| ToG | 82.6 | 68.5 | 45.7 |
| GCR | **92.2** | **75.8** | 49.9 |
| RoG | 85.7 | 62.6 | 57.6 |
| SubgraphRAG (200) | 90.5 | 63.5 | **61.6** |

On the other hand, this cross-dataset comparison can also be interpreted in the light of question complexity, as analyzed in Section 5: WebQSP and CWQ mostly consist of questions requiring simple reasoning paths, a setting where we also observed that GCR performs well (Figure 2). Its performance, however, degrades on questions with multiple seeds or requiring longer reasoning paths, as in most of the questions in GTSQA, leading to an overall worse ranking on our dataset. A similar argument can be applied to PoG, which, as we observed in Section 5, excels on 1-hop questions. On the other hand, RoG has a poorer performance on such simple questions (Figure 2), but it then outperforms other models on more complicated queries, explaining why it achieves a much better ranking on GTSQA.

In Figure 12 we provide additional results for the analysis in Section 6, quantifying improvements across different metrics for KG-RAG models trained on the ground-truth subgraphs in GTSQA, compared to training on shortest paths between seed and answer nodes, as conventionally done.

### C.3.1 Different reasoning LLMs

We repeat the main analysis of Section 5, but using the older GPT-4o-mini (OpenAI, 2025) as the model performing the final reasoning on the KG-retrieved subgraphs, to understand how the capabilities of the LLM influence Hits (EM). While all models score significantly lower than with GPT-5-mini (Table 8), the relative ranking remains the same as observed before, and no different patterns or behaviors arise when looking at individual graph isomorphism types (Figure 15) or at different generalization abilities required to answer (Figure 16).

We notice that SubgraphRAG is the model benefiting the most from using a more recent LLM to reason over the retrieved subgraph. This appears to be due to an improved utilization of long context, filtering out the noise in the augmented prompt when presented with a large number of retrieved triples. Indeed, while with GPT-4o-mini we observe a gap between model response quality as measured by Hits (EM) and recall of ground-truth triples when the number of retrieved triples is increased (Figure 14), GPT-5-mini exhibits continued improvements in Hits (EM) up to 500 triples, matching closely the steady increase in GT triple recall. This is reflected, across all models, in an even stronger linear correlation between Hits (EM) and the recall of ground-truth triples, with Pearson correlation coefficient increasing from $r = 0.95$ ($p = 1.3e{-}7$) for GPT-4o-mini to $r = 0.97$ ($p = 1.5e{-}8$) for GPT-5-mini (Figure 13).

Finally, we also repeat the analysis of Section 6 with GPT-4o-mini: as reported in Table 9, we still observe significant improvements (up to +20%) in the final predictive capabilities of the KG-augmented solutions when the KG retriever is trained on ground-truth paths, rather than shortest paths.

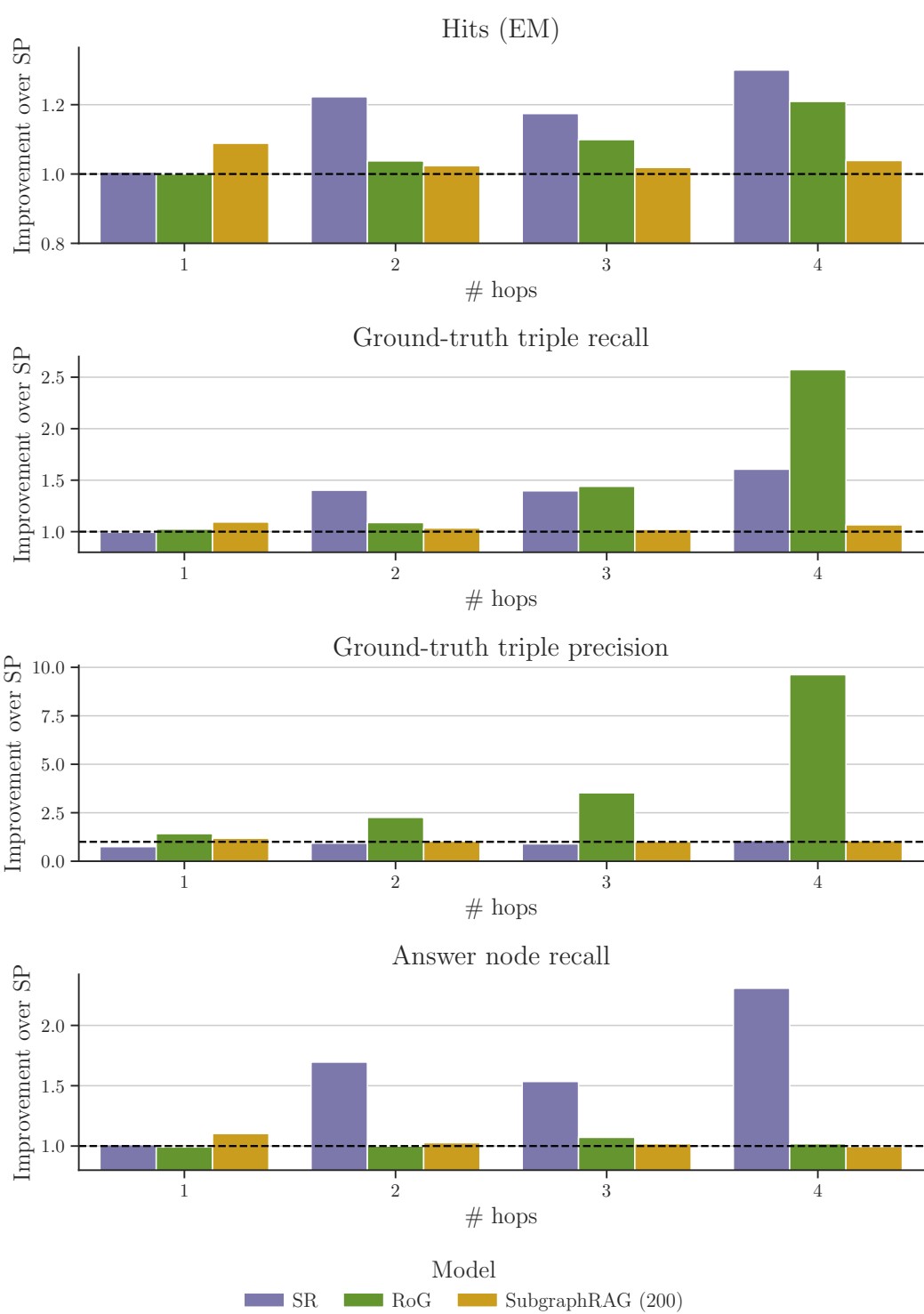

Figure 12: Comparison of predictive statistics when training models on ground-truth (GT) answer subgraphs compared to training on shortest paths (SP), for questions requiring different number of hops. The y-axis measures the GT/SP ratio of the averages of the statistic on the test set (over three distinct training runs); if the ratio is above the dashed line, training on the ground-truth subgraph is better.

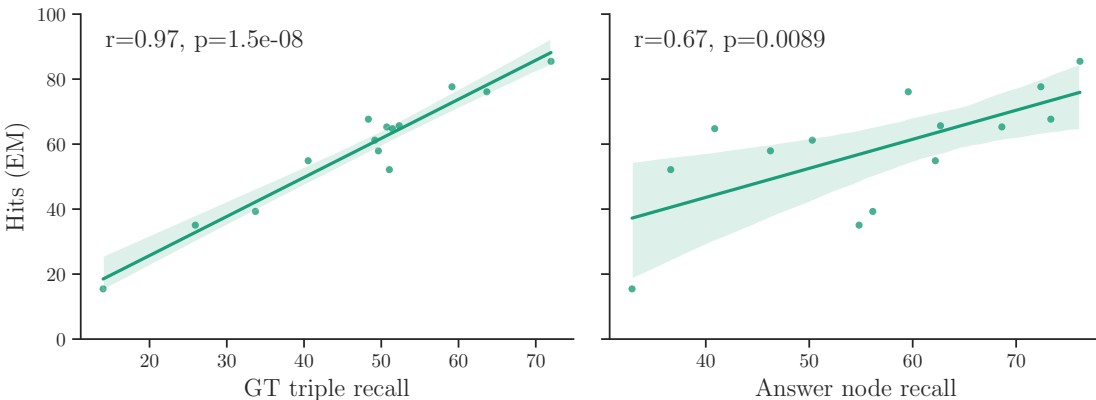

Figure 13: Correlation between Hits (EM) and recall of ground-truth triples (*left*) and of answer nodes (*right*), for questions in the test set of GTSQA. Each dot represents the average performance of the evaluated KG-RAG models on a different graph isomorphism type. While both recall variables have a positive linear correlation with predictive performance, recall of ground-truth triples is a significantly stronger predictor.

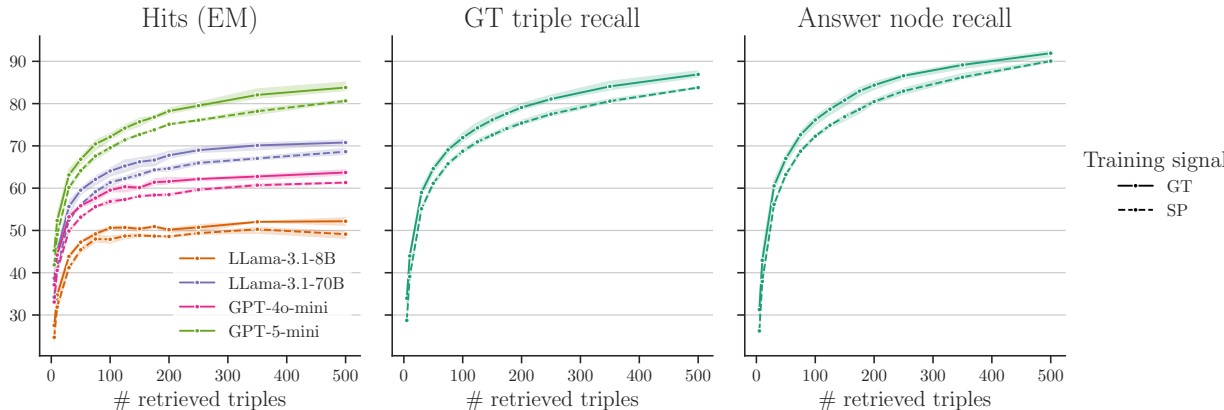

Figure 14: Performance of SubgraphRAG (with different LLMs doing the final reasoning) on the test set of GTSQA, as we increase the size of the retrieved subgraph. More capable LLMs are less sensitive on the noise in the retrieved data, with Hits (EM) tracking more closely the increase in recall of ground-truth triples and answer nodes. Across all LLM and subgraph sizes, training on the ground-truth answer subgraphs (GT) outperforms training on shortest path triples (SP).

Table 8: Benchmark of KG-RAG models on GTSQA with GPT-4o-mini as final reasoning model.

| Category | Model | EM | | Ground-truth triples | | | Answer nodes | | # triples |
| | | Hits | Recall | Recall | Precision | F1 | Hits | Recall | |
|---|---|---|---|---|---|---|---|---|---|
| KG agent | PoG | 32.92 | 31.60 | 31.95 | 27.52 | 27.03 | 31.81 | 30.54 | 7.09 |
| | ToG | 45.68 | 44.47 | 35.50 | 6.45 | 10.50 | 41.55 | 40.97 | 9.17 |
| Path-based | SR | 40.63 | 39.10 | 30.22 | 3.44 | 5.69 | 50.25 | 49.39 | 72.94 |
| | GCR | 49.91 | 48.25 | 40.71 | 27.21 | 29.82 | 47.11 | 45.54 | 6.54 |
| | RoG | 57.58 | 55.78 | 54.69 | 24.04 | 27.00 | 72.91 | 71.84 | 72.31 |
| All-at-once | SubgraphRAG (200) | 61.59 | 58.62 | 79.09 | 1.29 | 2.53 | 85.33 | 84.36 | 199.61 |

Table 9: Improvements in predictive accuracy of models trained on ground-truth answer subgraphs (GT), compared to models trained on the shortest paths between seed and answer nodes (SP), when final reasoning is performed by GPT-4o-mini. Experimental setting and reporting format as in Table 4.

| | Hits (EM) | |
|---|---|---|
| **Model** | SP | GT |
| SR | 33.95 | 40.63 (+20%; 4.5e-12) |
| RoG | 53.08 | 57.58 (+8%; 3.9e-6) |
| SubgraphRAG | 58.47 | 61.59 (+5%; 8.3e-4) |

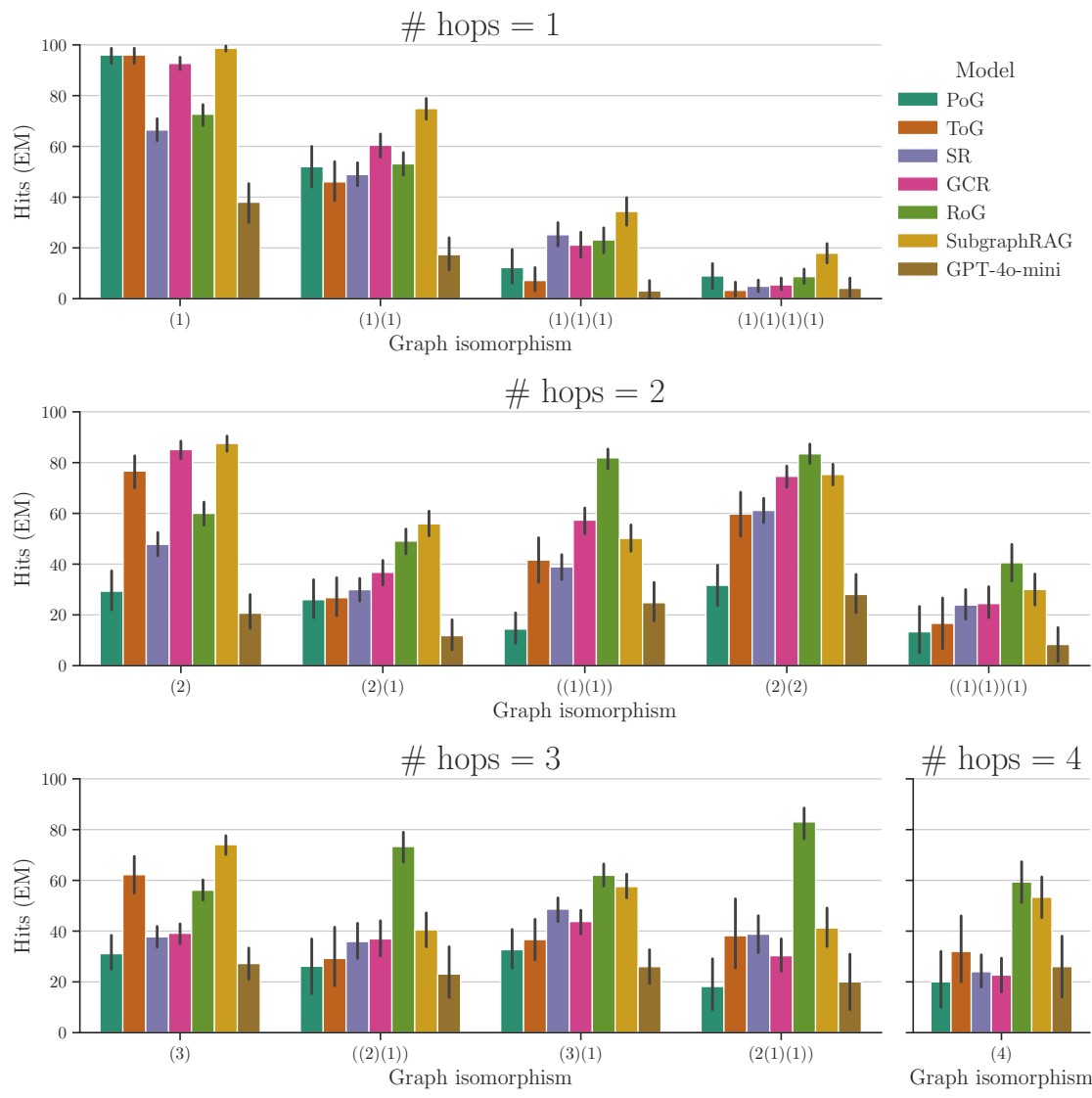

Figure 15: Hits (EM) of KG-RAG models on different graph isomorphism types, compared to the baseline (GPT-4o-mini, no RAG).

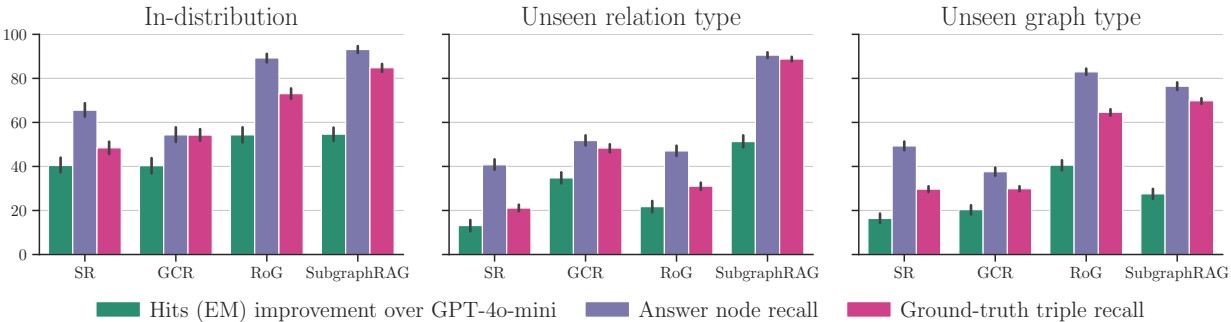

Figure 16: Generalization abilities of trainable KG-RAG models. We measure EM performance in terms of difference with the EM of the baseline (GPT-4o-mini, no RAG).

## D    Case Study of Retrievers' Failures

By presenting explicit examples, we discuss some common failure cases of different KG-RAG models, identified through the analysis in Section 5 and manual exploration of the experimental data. For all examples, the provided statistics are averaged over three runs of the models.

**Multi-seed questions**    As remarked in Section 5, all evaluated models perform poorly when the question requires expanding and combining reasoning paths from 3 or more seed entities. RoG and SubgraphRAG (and, to a lesser extent, SR) show better performance than other methods, as they typically retrieve larger sets of triples, which often mitigates the issue.

*Question*: What common genre do Johan Georg Schwartze and the painting by Alexander Andreyevich Ivanov in the Russian Museum share?

*Seed entities*: Johan Georg Schwartze, Alexander Andreyevich Ivanov, Russian Museum.

*Answers*: landscape art.

*Ground-truth answer subgraph*: (Johan Georg Schwartze; genre; landscape art), (Water and Stones near Palazzuola; creator; Alexander Andreyevich Ivanov), (Water and Stones near Palazzuola; collection; Russian Museum), (Water and Stones near Palazzuola; genre; landscape art).

*Graph isomorphism*: ((1)(1))(1).

*Generalization type*: unseen graph type.

| Model | Hits (EM) | Ground-truth triples | | | Answer nodes | # triples |
|---|---|---|---|---|---|---|
| | | Recall | Precision | F1 | Recall | |
| GPT-5-mini | 0.00 | - | - | - | - | - |
| PoG | 0.00 | 25.00 | 14.29 | 18.18 | 100.00 | 7.00 |
| ToG | 0.00 | 0.00 | 0.00 | 0.00 | 0.00 | 0.00 |
| SR | 0.00 | 33.33 | 1.50 | 2.82 | 100.00 | 107.67 |
| RoG | 100.00 | 100.00 | 5.80 | 10.96 | 100.00 | 69.00 |
| GCR | 0.00 | 25.00 | 41.67 | 30.56 | 100.00 | 2.67 |
| SubgraphRAG | 0.00 | 75.00 | 1.50 | 2.94 | 100.00 | 200.00 |

- **GPT-5-mini**: without any augmentations, the answer provided by the LLM is "They are both history paintings (historical/religious painting)". No information on the painting is provided, making hard to assess where the hallucination originates from.

- **PoG**: the model is able to retrieve the ground-truth triple (Johan Georg Schwartze; genre; landscape art). It also expands the search from the other two seed entities, retrieving multiple 1-hop paths from each of them, but it is unable to identify the painting that satisfies at the same time both the conditions in the question.

- **ToG**: the model proceeds to iteratively expand the search from only one of the seed nodes (Johan Georg Schwartze). When the maximum depth of exploration is reached, since not enough information to answer the question has been collected, ToG reverts to only answering using the LLM knowledge (Appendix C.2).

- **SR**: the unique relation paths predicted by the model are: (location, genre); (location, genre, field of work); (genre, depicts); (located in the administrative territorial entity). When looking for (possibly partial) realizations of these relation paths in the KG (in any direction), starting from the seed entities, the model retrieves the ground-truth triple (Johan Georg Schwartze; genre; landscape art). It also retrieves (Water and Stones near Palazzuola; location; Russian Museum), (Water and Stones near Palazzuola; genre; landscape art), through the metapath (location; genre) starting from the seed node "Russian Museum" (a parallel path, still valid, compared to the one in the ground-truth answer subgraph). However, no relation types pertaining to painting authorship are predicted.

- **RoG**: the model predicts much better relation paths than SR, namely: (genre); (inverse of: collection, genre); (inverse of: creator, genre); (collection, genre). As a result, all ground-truth triples are correctly retrieved (together with $\sim 65$ more).

- **GCR**: as we observe frequently in results from this method, GCR focuses on a single seed entity when decoding graph-constrained paths. Here, after de-duplication of the outputs, we find ourself with only two paths: *Johan Georg Schwartze $\rightarrow$ genre $\rightarrow$ landscape art*; *Johan Georg Schwartze $\rightarrow$ genre $\rightarrow$ portrait*. Thus, only one triple in the ground-truth answer subgraph is retrieved, namely (Johan Georg Schwartze; genre; landscape art).

- **SubgraphRAG**: the top-200 triples retrieved by the model contain three ground-truth edges, namely: (Johan Georg Schwartze; genre; landscape art), (Water and Stones near Palazzuola; creator; Alexander Andreyevich Ivanov), (Water and Stones near Palazzuola; collection; Russian Museum). The model is therefore able to identify the painting satisfying the two conditions in the query, but not to make the additional hop from it to the answer node. As highlighted in Section 5, this is likely due to the poor generalization abilities of SubgraphRAG to new graph isomorphisms, as no questions requiring additional projections after the intersection of paths from different seed entities have been observed during training.

**Multi-hop questions**   Questions requiring more than two hops, even in the presence of a single seed entity, pose significant challenges to most KG retrievers.

*Question*: Who is the lyricist of the national anthem for the country that is home to the Nauru Reed Warbler?

*Seed entities*: Nauru Reed Warbler.

*Answers*: Margaret Hendrie.

*Ground-truth answer subgraph*: (Nauru Reed Warbler; endemic to; Nauru), (Nauru; anthem; Nauru Bwiema), (Nauru Bwiema; lyrics by; Margaret Hendrie).

*Graph isomorphism*: (3).

*Generalization type*: unseen relation type (relation "endemic to" not included in the train set).

| Model | Hits (EM) | Ground-truth triples | | | Answer nodes | |
| | | Recall | Precision | F1 | Recall | # triples |
|---|---|---|---|---|---|---|
| GPT-5-mini | 0.00 | - | - | - | - | - |
| PoG | 0.00 | 33.33 | 100.00 | 50.00 | 0.00 | 1.00 |
| ToG | 100.00 | 100.00 | 10.71 | 19.35 | 100.00 | 28.00 |
| SR | 66.67 | 66.67 | 10.25 | 17.67 | 66.67 | 33.00 |
| RoG | 0.00 | 0.00 | 0.00 | 0.00 | 0.00 | 0.00 |
| GCR | 0.00 | 66.67 | 40.00 | 50.00 | 0.00 | 5.00 |
| SubgraphRAG (200) | 100.00 | 100.00 | 1.50 | 2.96 | 100.00 | 200.00 |

- **GPT-5-mini**: the answer provided by the LLM is "The Nauru Reed Warbler originates from Nauru. The national anthem of Nauru is 'Nauru Bwiema,' and the lyricist is the former president Hammer DeRoburt". The model is therefore able to correctly identify the nation and the anthem, but it hallucinates the identity of the lyricist.

- **PoG**: the model correctly retrieves the first ground-truth triple (Nauru Reed Warbler; endemic to; Nauru). However, the agent then decides to prematurely stop the search and directly generate the answer (this is a common behavior of PoG on multi-hop questions, as observed in Section 5).

- **ToG**: in a scenario with a single seed entity, the KG retriever is able to effectively expand the search and fetch a set of $\sim 30$ triples containing all the ground-truth ones.

- **SR**: there are only a few relation types generating from the seed entity, hence the model is able to identify sensible relation paths. In two tries out of three, it predicts: (IUCN conservation status, IUCN conservation status, depicts); (endemic to, anthem); (endemic to, anthem, lyrics by); (endemic to, anthem, composer). This leads to retrieving all the ground-truth triples.

- **RoG**: the relation paths predicted by RoG are (country of origin, anthem, lyrics by); (country of citizenship, anthem, lyrics by); (inverse of: has part, lyrics by); (inverse of: native bird, anthem, lyrics by). While the two ground-truth relation types seen during training ("anthem", "lyrics by") are consistently predicted, the model is unable to come up with good suggestions for the unseen relation type "endemic to", proposing instead known relation types with a similar meaning (e.g., "country of origin") or hallucinating nonexistent relation types, such as "native bird". None of the proposed relation paths is realized in the KG starting from the seed entity, hence the set of retrieved triples is empty.

- **GCR**: the graph-constrained decoding from the seed entity leads to identifying the following paths: *Nauru Reed Warbler → taxon rank → species → inverse of: taxon rank → Nauru Reed Warbler*; *Nauru Reed Warbler → endemic to → Nauru → anthem → Nauru Bwiema*; *Nauru Reed Warbler → IUCN conservation status → Vulnerable → inverse of: IUCN conservation status → Nauru Reed Warbler*; *Reed Warbler → parent taxon → Acrocephalus → inverse of: parent taxon → Nauru Reed Warbler*. While many paths reverse to the seed entity, one of them correctly predicts the first two hops. Since we stop the search at depth two, due to the costs of the implementation (Appendix C.2), this is the best that the model can achieve.

- **SubgraphRAG**: the three ground-truth triples are ranked 5th, 8th, 9th with respect to the relevance scores assigned by the model, hence they are all consistently retrieved.

# E  SynthKGQA on Domain-Specific KGs

As stressed in Section 3, the SynthKGQA framework can be applied to any custom Knowledge Graph, including domain-specific ones. To showcase this flexibility, we conduct a small-scale experiment on Hetionet (Himmelstein et al., 2017), a biomedical Knowledge Graph used to train ML models that support practitioners in the drug discovery process. The 45,000 nodes in Hetionet represent genes, diseases, biological pathways, compounds, etc. and KG edges encode experimentally-validated interactions among them (24 different relation types).

When generating 1000 KGQA datapoints with SynthKGQA, we observe an acceptance rate (after candidate validation via SPARQL submission, Appendix A) similar to the one achieved on Wikidata during the generation of GTSQA, namely between 10% and 25%, depending on the complexity of answer subgraphs. As we did for GTSQA, we performed a final filtering step by tasking GPT-4o-mini to answer the accepted questions when augmenting the prompt with the ground-truth answer subgraph selected by the LLM and excluding datapoints with incorrect answers. The percentage of questions failing this test is slightly above what we observed for GTSQA, but still extremely low, at 1.17%.

A few randomly examples (in compact form) from the final set of questions are shown below.

---

**Example**

```
"paraphrased_question":  "What compound decreases the EBP gene expression while also leading
to anxiety and elevated blood creatinine levels as side effects?",

"answer_node":  "Sirolimus",

"answer_subgraph":  [["Sirolimus", "Compound - downregulates - Gene", "EBP"], ["Sirolimus",
"Compound - causes - Side Effect", "Anxiety"], ["Sirolimus", "Compound - causes - Side
Effect", "Blood creatinine increased"]],

"all_answers_hetionet":  ["Pentamidine", "Sirolimus", "Temsirolimus"],

"n_hops":  1,

"graph_isomorphism":  "(1)(1)(1)",
```

---

**Example**

```
"paraphrased_question":  "Which gene, regulated by ZNF433, also covaries with a gene involved
in the sphingolipid metabolism?",

"answer_node":  "PIGB",

"answer_subgraph":  [["ZNF433", "Gene > regulates > Gene", "PIGB"], ["CERS1", "Gene -
covaries - Gene", "PIGB"], ["CERS1", "Gene - participates - Biological Process", "sphingolipid
metabolic process"],

"all_answers_hetionet":  ["PIGB"],

"n_hops":  2,

"graph_isomorphism":  "(2)(1)",
```

---

**Example**

```
"paraphrased_question":  "Which biological process involves a gene that is regulated by one
known to participate in the repression of the transcription factor binding by RNA polymerase
II?",

"answer_node":  "response to hyperoxia",

"answer_subgraph":  [["TCF7L2", "Gene - participates - Molecular Function", "RNA polymerase
II repressing transcription factor binding"], ["TCF7L2", "Gene > regulates > Gene", "BNIP3"],
["BNIP3", "Gene - participates - Biological Process", "response to hyperoxia"]],

"all_answers_hetionet":  ["response to hyperoxia", ...],

"n_hops":  3,

"graph_isomorphism":  "(3)",
```

---

## F   Comparison of SynthKGQA with Concurrent Frameworks

With the latest advancements in LLM reasoning abilities, strong interest has arisen in using generative AI to create synthetic datasets for a variety of applications (Long et al., 2024). Two recent works (Dammu et al., 2025; Zhang et al., 2025) have proposed pipelines that share similarities with the one outlined in Section 3 to create synthetic datasets for KGQA. Here, we discuss the differences of these methodologies in detail.

- Dynamic-KGQA (Dammu et al., 2025) is designed to build (question, answer subgraph) pairs on the fly using LLMs, starting from compact seed subgraphs in YAGO 4.5 (Suchanek et al., 2024) that group triples around a common theme. However, as observed in Zhang et al. (2025), the generation pipeline is still prone to hallucinations, with factual correctness of questions estimated to be not better than previous benchmarks ($\sim 45\%$). In fact, no SPARQL (or other logical) query is generated and executed on the KG: the LLM is tasked to judge whether the proposed answer subgraph captures the correct reasoning paths from seed to answer nodes, resulting in higher likelihood of incorrect data and hallucinations, compared to our validation approach based on SPARQL queries.

- KGQAGen (Zhang et al., 2025) proposes a more cost-efficient validation pipeline, based on SPARQL query execution and iterative revisions of incorrect queries (while we directly discard datapoints where the SPARQL query does not execute, or returns incompatible results with the LLM-generated ones). However, it does not keep track of the seed entities mentioned in the natural-language question, only providing the node from which the seed graph is constructed (equivalent to our entity $s$ in Algorithm 1). Through manual inspection we find that, in the majority (70.8%) of questions in the KGQAGen-10k test set, such "seed node" actually coincides with one of the answer nodes. This highlights a strong bias in the way KGQAGen constructs datapoints. Moreover, since such seed node is consistently used as a ground-truth seed entity (hence, the starting point of the KG retrieval) for the KG-augmented LLMs evaluated in the paper, this also leads to major information leakage and calls into question the benchmarks reported in Zhang et al. (2025). The datasets presented in the paper, KGQAGen-10k, contains 10,787 questions with a random 80/10/10 train/validation/test split (while we carefully curate the split of GTSQA to test zero-shot generalization abilities of models), constructed from Wikidata, with seed entities from a set of 16,000 Wikipedia's Vital Articles[4]. However, since no analysis of the isomorphism type of the ground-truth answer subgraphs is conducted (in particular, to avoid cycles), we observe a high occurrence of circular questions, where the answer is mentioned verbatim in the text of the question (9.4% of questions in the test set).

While both Dynamic-KGQA and KGQAGen are presented as multi-hop datasets, neither paper includes detailed statistics on the distribution of structures for their respective ground-truth answer subgraphs, making hard to actually evaluate the degree of question complexity in these datasets. On the other hand, our framework makes this very simple, using the metrics provided by the classification of graph isomorphism (Appendix A.2). Similarly, while Dynamic-KGQA uses an LLM as-a-judge to assess the presence of redundant information in the question, our approach based on decomposition of SPARQL queries and identification of minimal subsets of seed entities sufficient to answer (Appendix A.3) uniquely implements an exact measure of minimality. Finally, only GTSQA ensures better reliability as a benchmark for KG-augmented LLMs by checking that the ground-truth subgraphs provided can indeed be utilized by an LLM to answer test questions correctly.

---

[4]https://en.wikipedia.org/wiki/Wikipedia:Vital_articles

