# OpenReview forum: "Ground-Truth Subgraphs for Better Training and Evaluation of Knowledge Graph Augmented LLMs"
_TMLR — Under review for TMLR_

### Review · Reviewer_GmpB · 2026-07-02

**Summary Of Contributions:**

The paper introduces SynthKGQA, an LLM-based pipeline for generating KGQA data with verified ground-truth answer subgraphs and SPARQL queries. The authors instantiate the pipeline on Wikidata and release GTSQA, a dataset of roughly 32K questions with dedicated splits for unseen graph types and unseen relation types. Their experiments evaluate several KG-RAG methods and show that current systems struggle on difficult multi-seed cases, that training with ground-truth subgraphs is more effective than shortest-path supervision, and that models trained on GTSQA show reasonable transfer to Mintaka.

**Audience:**

Yes

**Audience Explanation:**

Yes. KG-RAG is an active area, and the lack of gold retrieval labels remains a practical obstacle for both training and evaluation. A dataset with ground-truth subgraphs and retrieval-level evaluation should be useful to researchers working on KG-augmented LLMs. The multi-seed failure analysis and the comparison between shortest-path and ground-truth supervision are also practically informative, beyond reporting another set of benchmark scores.

**Claims And Evidence:**

Yes

**Claims Explanation:**

Most of the central claims are supported by the experiments. The shortcut and parallel-path analysis gives a convincing explanation for why shortest paths can be a weak training target, and the ground-truth versus shortest-path comparison in Table 4 is clear and accompanied by significance testing. The main benchmark is also reasonably broad, covering multiple model families, subgraph-level metrics, and breakdowns by graph type and OOD setting.
Some claims are slightly stronger than the evidence warrants. Test questions are filtered to cases that GPT-4o-mini can answer with the gold subgraph, which may make the benchmark easier than the unfiltered generated set. Question naturalness is illustrated mainly through examples rather than a human evaluation. In addition, retrieval is performed on ogbl-wikikg2 rather than live Wikidata, so claims about an "up-to-date" KG should be phrased more carefully. Transfer is evaluated only on Mintaka. These issues do not overturn the main findings, but they should be addressed or scoped more explicitly.

**Requested Changes:**

Please analyze the test-set filtering step more carefully. Removing samples that GPT-4o-mini cannot solve with the gold subgraph is reasonable as a sanity check, but 0.47% is still worth characterizing. The paper should report what kinds of examples are removed and whether they differ systematically in hop count, graph type, or relation type.
Be explicit that benchmark retrieval uses ogbl-wikikg2 (2015 dump), while SPARQL validation uses current Wikidata. Right now this is easy to misread.
Add a small human check of question quality, even on 100–200 randomly sampled examples. The check should cover fluency, ambiguity, and whether the question matches the intended SPARQL logic.

Report generation cost and acceptance rate in more detail. An ablation with an open model would also help with reproducibility.
Transfer results on one additional human-curated dataset would make the external validation stronger.
A short discussion of the limitations introduced by tree-only subgraphs and provided seed entities would help scope the claims.
A direct comparison with KGQAGen under the same retrieval setup would strengthen the positioning against concurrent work.

---

### Review · Reviewer_E4Vn · 2026-07-05

**Summary Of Contributions:**

The paper addresses a real gap in KGQA: existing benchmarks (WebQSP, CWQ, GrailQA) lack the **ground-truth answer subgraph** — the exact triples needed to answer a question — so retrievers cannot be evaluated in isolation and must be trained on *shortest paths* as a proxy. It contributes:

1. **SynthKGQA**, a KG-agnostic framework where an LLM generates a question together with a **SPARQL query** that is *executed against the KG to validate the datapoint and discard hallucinations* (Section 3, Figure 1), plus per-datapoint isomorphism type, minimality, and a paraphrase.
2. **GTSQA**, 32,099 Wikidata-grounded questions with 27 subgraph isomorphism types, up to 5 hops/seeds, with a split designed for zero-shot generalization to unseen structures and relations (Section 4, Tables 1, 5).
3. A **benchmark and fine-grained analysis** of SOTA KG-RAG models (Section 5, Table 2), and the central claim, quantified for the first time, that **ground-truth subgraphs are a better training signal than shortest paths** (Section 6, Table 4).

**Key strengths.** SPARQL-execution validation is a principled hallucination filter and a genuine improvement over concurrent pipelines (Appendix F). The analysis is unusually granular and exposes a hidden failure mode (multi-seed intersection questions). The GT-vs-SP training claim is a controlled experiment across three architectures, mechanistically grounded in the low GT/SP overlap (Table 6, Figure 9). The authors release standardized retrieval graphs (Appendix C.1) and show generality (Hetionet, Appendix E; Mintaka transfer, Table 3).

**Key weaknesses** (verified against the text; details in Requested Changes). Dataset quality rests entirely on automated self-consistency checks with **no human evaluation** and an unvalidated LLM paraphrasing step. The **Table 4 significance tests** omit their unit of analysis and report p-values incompatible with a 3-vs-3 run test. Several headline numbers **overstate the fair metric**: EM gains are modest for strong retrievers (+4% SubgraphRAG), "+141% precision" holds for RoG only (SR precision *drops*), and "up to 30%" is a 4-hop slice (avg hops = 2.02). Reproducibility is limited by closed models, and cross-model retrieval metrics are confounded by retrieved-subgraph size.

**Additional Comments:**

No comments.

**Audience:**

Yes

**Audience Explanation:**

Yes. In the active KG-RAG area, the paper offers builders a training resource with true supervision targets and an actionable finding (train on ground-truth subgraphs, not shortest paths), and offers the benchmarking community a dataset that isolates retriever quality and exposes a hidden failure mode (multi-seed intersection questions). The ranking reshuffle vs. WebQSP/CWQ (Table 7) would itself prompt re-examination of established conclusions, and the KG-agnostic framework (Hetionet) extends interest to domain-specific KGs.

**Broader Impact Concerns:**

I have no significant ethical concerns, and I do not think a Broader Impact Statement is strictly required for acceptance. Two minor points the authors could optionally address:

1. **Propagation of synthetic errors.** Because the dataset is LLM-generated, any residual hallucinations that survive SPARQL validation (see Requested Change #1) could propagate into models trained on GTSQA and into downstream factuality claims. The SPARQL-execution filter mitigates this well, but a sentence acknowledging the residual risk and pointing to the human-evaluation results would be appropriate.

2. **Factual currency.** Answers are tied to a specific Wikidata dump; facts change over time, so the dataset can encode assertions that later become incorrect. The paper's design (re-runnable SPARQL to regenerate the dataset) already addresses this, and it is worth stating explicitly as a maintenance/impact note.

**Claims And Evidence:**

Yes

**Claims Explanation:**

I answer Yes because the paper's *central* claims rest on appropriately controlled and mechanistically grounded evidence. The claim that ground-truth subgraphs beat shortest paths as a *training* target is a controlled experiment varying only the supervision signal across three architectures over three runs (Section 6, Table 4, Figure 12), and it is explained rather than merely observed: for 4-hop questions only 13.3% of shortest-path triples lie in the GT subgraph and 72.5% of GT triples are off any minimal path (Table 6, Figure 9). Dataset difficulty, the multi-seed failure mode, and the ranking differences vs. WebQSP/CWQ are consistent across many models (Tables 2, 7; Figures 2, 10, 11), and generality is each backed by a dedicated experiment (Mintaka, Table 3; Hetionet, Appendix E).


That said, several claims are phrased **more strongly than the evidence supports** and should be backed by more evidence or calibrated (all verified against the submission):

1. **The training benefit is real but modest on the fair metric, and uneven.** EM (Hits) is the metric not defined by the GT labels, and there the improvement is +14% (SR), +5% (RoG), and only **+4% (SubgraphRAG)** — the strongest retriever (Table 4). The large "+141% precision" applies to **RoG only**; for SR, GT-triple precision actually **decreases** (3.84 → 3.44, p=0.99, not significant). The abstract's "up to 30%" corresponds to SR at 4 hops (Figure 12), a slice with few test questions (avg hops = 2.02, Table 1). The direction of the result is well supported; the magnitude framing is not, and should be tied to EM and stated per-architecture.

2. **Dataset quality is asserted, not measured.** Validation is by SPARQL execution plus a GPT-4o-mini answerability filter (0.47% failure, Section 4). But the SPARQL is written by the same model that wrote the question, so a semantic mismatch between question and subgraph is only caught if it also flips the answer entity; the answerability filter uses a same-family model; and the LLM paraphrasing step (Appendix A.4) is never validated (it is only observed to "sound natural"). The 0.47% figure may partly reflect self-consistency rather than correctness.

3. **The Table 4 significance tests are not interpretable as reported.** The caption states a two-sample one-sided t-test on "the mean of distribution of SP/GT scores," with results "averaged over three distinct runs." A 3-vs-3 run-level test cannot yield p ≈ 6.2e-22. The unit of analysis (per-question vs. per-run) must be stated and its independence assumption justified.

These are correctable via clarification, calibration, and one additional study, so I do not consider them grounds to answer No — but they are the conditions on which my recommendation rests (Requested Changes #1–#3).

**Requested Changes:**

**Critical (required for my recommendation to accept):**

1. **Human evaluation of dataset quality.** Add a study on a random sample (≥150–200 datapoints, stratified by isomorphism type and hop count) with ≥2 annotators rating: (a) semantic faithfulness of the natural-language question to the ground-truth subgraph/SPARQL, (b) fluency/naturalness of the *paraphrased* question, and (c) answer correctness. Report inter-annotator agreement. Also state explicitly whether the GPT-4o-mini answerability filter is applied to the original or the paraphrased question. This is the single most important gap: the "high-quality, human-free" claim currently rests on self-consistency between components produced by the same model family.

2. **Clarify and, if needed, correct the statistics in Table 4 (and Table 9).** State the exact test, the sample it is computed over, and the number of samples. If the reported p-values derive from per-question aggregation, say so and justify independence; if from run-level means, the magnitudes need re-checking. Justify the use of a one-sided test. Without this, the significance stars cannot be interpreted.

3. **Calibrate the headline claims to the fair metric.** (a) Anchor the training-benefit claims to EM and report them per-architecture, making clear the effect is modest for the strongest retriever (+4% SubgraphRAG). (b) Note that the precision improvement is not universal (RoG only; SR decreases). (c) Qualify "up to 30%" as a 4-hop result and report the number of 4-hop test questions supporting it. (d) Since GT-triple recall/precision are defined by the GT labels the model is trained on, foreground EM as the primary evidence and treat GT-triple recall/precision as secondary.

**Would strengthen the work (not blocking):**

4. **Quantify the question-specific-graph bias (Appendix C.1).** Re-adding pruned ground-truth edges (which you note "unfairly bias[es] retrievers") plus same-metapath confounders leaves a *net* bias that is unquantified. Report how often GT edges are re-added and an ablation showing how sensitive the Table 2 rankings are. (Releasing standardized graphs is the right fix for prior work's reproducibility problem; this is only about the residual bias.)

5. **Reproducibility of the benchmark reasoner.** Report Table 2 with at least one strong *open-weight* model as the fixed final reasoner (in addition to the proprietary ones) and pin exact version/date for all closed models, decoupling the ranking from a closed endpoint. (Table 8's GPT-4o-mini variant is a start but still proprietary.)

6. **Fairness of GCR in the leaderboard.** GCR is restricted to 2-hop and excluded from Section 6, yet kept in Tables 2/7. Flag there that its GTSQA numbers are lower-bounded by this restriction.

7. **Size-controlled retrieval comparison.** Cross-model recall/precision/F1 (Table 2) are confounded by retrieved-subgraph size (3.72–199.61 triples). Add a recall-at-matched-#triples (Pareto) view across models, analogous to the SubgraphRAG sweep in Figure 14.

8. **Secondary answer metric.** EM can under-credit pure-LLM baselines that know the answer but phrase it differently; add an alias-aware or LLM-judge metric for the LLM-only rows to confirm the "challenging benchmark" claim is not partly an EM artifact.

9. **Head-to-head vs. concurrent datasets.** The KGQAGen/Dynamic-KGQA comparison (Appendix F) is qualitative; one head-to-head experiment (same retriever trained on GTSQA vs. KGQAGen, compared on transfer) would turn "more reliable" into a demonstration.

10. **Scope statement.** State in the main text that GTSQA is restricted to tree-structured, entity-valued *conjunctive* queries (aggregation/negation/comparatives are appendix-only extensions), and report the fraction of generated questions discarded for being non-tree — a genuine trade-off (structurally richer but operator-poorer than CWQ/GrailQA).

11. **Complexity measure.** Note that the isomorphism type captures only structural complexity, not semantic difficulty (relation ambiguity, entity popularity).

12. **Minor/presentation.**
   - Figure 11 is very dense; consider splitting or enlarging.
   - Table 5 mixes Dutt et al. Iso-IDs and N/A rows; a one-line note on why some types have no prior ID would help.
   - Define "minimality" intuitively at first use in the main text (Section 3), not only in Appendix A.3.
   - Table 2: clarify in the caption that "# triples" is the average size of the retrieved subgraph.
   - State the exact Wikidata dump date in the main text (currently only in a footnote) since it governs answer correctness.

---

### Review · Reviewer_9PWr · 2026-07-16

**Summary Of Contributions:**

This paper presents SynthKGQA, an LLM-based framework for automatically constructing knowledge graph question answering (KGQA) datasets with verified ground-truth answer subgraphs and executable SPARQL queries. Based on this framework, the authors introduce GTSQA, a new benchmark designed to evaluate KG retrieval under multi-hop, multi-seed, and zero-shot generalization settings.

Beyond introducing a new dataset, the paper argues that existing KGQA benchmarks lack explicit retrieval supervision, making it difficult to fairly evaluate or train KG retrievers. The proposed framework addresses this issue by providing complete ground-truth retrieval targets. Extensive experiments benchmark several representative KG-RAG systems, analyze their retrieval behavior across different graph structures, and demonstrate that training retrievers with ground-truth subgraphs consistently outperforms the commonly adopted shortest-path supervision.

**Audience:**

Yes

**Audience Explanation:**

Yes.

Knowledge graph augmented LLMs and graph retrieval have become increasingly important topics in retrieval-augmented generation. One current bottleneck in this area is the lack of reliable datasets for evaluating and training KG retrievers. This work addresses that gap by introducing both a new benchmark and a reproducible data generation framework.

Beyond the released dataset itself, the paper provides several practical insights into retrieval supervision, evaluation methodology, and failure analysis of existing KG-RAG systems. These findings are likely to be valuable for researchers working on KGQA, graph retrieval, and retrieval-augmented LLMs more broadly.

**Broader Impact Concerns:**

I do not identify major ethical concerns requiring additional discussion beyond standard considerations.

**Claims And Evidence:**

Yes

**Claims Explanation:**

Overall, the major claims are well supported by the presented evidence.

The paper includes extensive experimental validation from multiple perspectives, including comparisons across representative KG-RAG models, retrieval quality analysis using ground-truth subgraphs, zero-shot generalization evaluation, cross-dataset transfer experiments, and ablation studies comparing shortest-path supervision with ground-truth supervision. These experiments consistently support the paper’s central claims regarding the value of explicit retrieval supervision.

In addition, the proposed data generation pipeline incorporates automatic validation via SPARQL execution and further sanity checks using LLM reasoning, which increases confidence in the correctness of the generated benchmark. While some conclusions naturally depend on synthetic data generated by the proposed pipeline, the empirical evidence is generally convincing and sufficiently comprehensive.

**Requested Changes:**

Minor revisions :

1. The paper would benefit from a clearer discussion of the computational cost of the SynthKGQA pipeline, including the number of LLM calls, SPARQL executions, and the approximate cost of generating datasets of different scales.
2. Although the paper briefly demonstrates applicability to a biomedical knowledge graph, a more extensive evaluation beyond Wikidata would further strengthen the generality claims.
3. It would be helpful to include a more explicit comparison between SynthKGQA and concurrent synthetic KGQA generation frameworks in the main paper rather than primarily in the appendix, especially highlighting differences in supervision signals and benchmark design.
4. Some discussion of potential biases introduced by LLM-generated questions (e.g., linguistic style, reasoning patterns, or distributional bias) would further improve the paper.